# Analysis of Forest Fragmentation and Connectivity Using Fractal Dimension and Succolarity

**Ion Andronache** 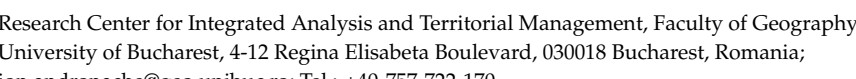

Research Center for Integrated Analysis and Territorial Management, Faculty of Geography, University of Bucharest, 4-12 Regina Elisabeta Boulevard, 030018 Bucharest, Romania; ion.andronache@geo.unibuc.ro; Tel.: +40-757-722-170

**Abstract:** Forests around the world, vital for ecological, economic, and social functions, are facing increasing threats such as deforestation and habitat fragmentation. This study introduces "succolarity" to analyse forest connectivity and fragmentation directionally in the Romanian Carpathians from 2000 to 2021. Besides traditional indices such as the fractal fragmentation index (*FFI*), the fractal fragmentation and disorder index (*FFDI*), the local connected fractal dimension (*LCFD*), and succolarity, two novel indices are proposed: potential succolarity and delta (Δ) succolarity, which provide nuanced insights into environmental changes and human interventions in forests worldwide. The succolarity tests showed invariance at the image scale and sensitivity to the obstacles in different image scenarios. The combined analysis of succolarity and fractal indices reveals dynamic patterns of connectivity, fragmentation, and spatial disturbance. The directional insights of succolarity highlight and enhance understanding of deforestation patterns. The *FFI* and *FFDI* show different levels of fragmentation across mountain groups, while the *LCFD* details local connectivity. The adaptability of the method makes it globally applicable, supporting biodiversity conservation and landscape management. In conclusion, the integration of succolarity and traditional fractal indices provides a robust methodology for the comprehensive analysis of forest fragmentation. This advancement not only supports the sustainable management of the Romanian Carpathians but also represents a versatile approach applicable to ecosystems worldwide, ensuring the long-term resilience of forested regions.

**Keywords:** succolarity; potential succolarity; delta succolarity; percolation; permeability; fragmentation; connectivity; fractal analysis; forests; deforestation; habitat; ecosystem

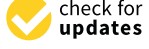



## 1. Introduction

Forests are widely recognised as crucial reservoirs of ecological, economic, and social benefits [1]. However, they are currently facing a growing number of threats, particularly habitat fragmentation and loss [2–4]. Reports by the Food and Agriculture Organization [5] indicate that an average of approximately 10 million hectares are lost annually between 2015 and 2020. Although forests play a vital role in mitigating climate change by absorbing an impressive 2 billion tons of carbon dioxide each year [6], deforestation rates have surged to alarming levels globally. This hinders the overall effectiveness of forests in acting as carbon sinks, thereby reducing their impact on climate change [7].

For instance, deforestation and habitat fragmentation over large areas are two interrelated processes that have significant impacts on forests, forest ecosystems, and biodiversity [8–11]. The natural space for numerous species of plants and animals that rely on extensive forest areas becomes smaller and isolated with habitat fragmentation [12], which complicates the movement and migration of these species, reducing their chances of reproduction and survival [13–15]. The fragmentation of habitats has negative effects on the connectivity between different fragments; thus, loss of genetic diversity can impede seeding and pollination, negatively affecting natural forest regeneration [16]. Additionally, fragmentation can impact the natural resources of the forest, such as water and soil resources [17,18].

To address these complex challenges effectively, a thorough and detailed analysis of habitat fragmentation is necessary not only for developing effective conservation strategies [19,20] but also for informing ecological restoration initiatives and promoting sustainable urban development planning [21–23]. Various methods are available for testing [24–31] or assessing forest fragmentation, including the weighted mean patch size [32], the isolation index [33], path density [34], the shape index [35], the connectivity metrics [36,37] or the patch fragmentation index [38].

Among these methods, fractal analysis is also included. Fractal indices are a method used to measure the fragmentation and complexity of a landscape or ecosystem, including forests [39]. They have the advantage of being easy to compute and scale invariant [40]. The analysis of fractal indices can assess the impact of environmental changes and human interventions on the fragmentation and connectivity of forest habitats [41]. These indices are based on concepts from mathematics, specifically nonlinear dynamics and fractal geometry, and can provide a different perspective on how a landscape or forest is fragmented. Various measures have been proposed to quantify the degree of fragmentation in a landscape, including the box-counting dimension [42–44], the pyramid dimension [45,46], lacunarity [47–49], the Minkowski dimension [50], the Higuchi dimension [51,52], the local connected fractal dimension [53], the fractal fragmentation indices [39,54], the generalized dimension [55], or generalised entropies [56].

From a fractal perspective, existing indices such as the fractal fragmentation index (*FFI*) [54], the fractal fragmentation and disorder index (*FFDI*) [39], and the local connected fractal dimension (*LCFD*) [53] quantify only one characteristic of forest structure—either fragmentation, disturbance, or connectivity—globally, without considering directionality. Although these indicators provide valuable results [39,54,57–62], they do not establish a direct link between fragmentation and connectivity. Succolarity may prove useful in overcoming this limitation. It has the advantage of directionality and only quantifies dynamically connected forest patches.

However, these measures may not provide detailed information on the directionality and specific obstacles that affect connectivity between fragmented patches.

This study aims to fill the gap by using an innovative approach that leverages the concept of succolarity. This study adapts and enhances the fractal measure of succolarity to evaluate how fragmentation affects connectivity in forest areas, considering directional barriers. Succolarity was originally designed to measure the permeability of fragmented images, specifically the degree to which "virtual water" can permeate a binary image [63–65]. In this context, white pixels represent obstacles, while black pixels represent pathways or connectivity for said virtual water. To analyse forests using succolarity, a fractal concept for quantifying percolation, i.e., tree cover, is defined as those areas where virtual water can percolate, similar to a lake (represented by black pixels). Areas without forests are considered obstacles that force the virtual water to bypass or be blocked (represented by white pixels), such as the space between some lakes. This study presents a novel method for evaluating the impact of fragmentation on connectivity in forested regions, taking into account directional barriers.

The results of our succolarity analysis [65] are compared with three existing fractal models: the fractal fragmentation index [54], the fractal fragmentation and disorder index [39], and the local connected fractal dimension [53].

In Sections 2.1–2.7, the different scenarios introduced for the fractal indices are described and provided essential support for ensuring the efficacy of the two new concepts proposed in this study—potential succolarity and Δ succolarity—in better understanding percolation expansion and forest landscape dynamics for the tree cover database employed. The advantages and limitations are discussed in Section 4 in some detail.

This research analyses images of forest areas in the central regions of 10 mountain groups in the Romanian Carpathians between 2000 and 2021 that are undergoing an intensive process of deforestation, both legal and illegal, for economic purposes such as timber production for export, the expansion of residential areas, or the establishment of

new tourist facilities [57,61,62,66]. The choice of this case study is driven by the significant reduction in bear habitats in the Carpathians, forcing bears to enter human settlements in search of food. Thus, quantifying the degree of fragmentation and connectivity, correlated with potential connectivity, could identify potential areas for habitat restoration [67] that would significantly reduce bear–human interactions, especially considering the protected status of bears [68–73].

This research presents a noteworthy methodological improvement in quantifying the effects of deforestation and forest loss in the Romanian Carpathians. A better comprehension of the dynamics of forested regions provides significant opportunities for preserving biodiversity and ensuring the long-term sustainability of these mountain ecosystems.

The use of succolarity can offer valuable insights into the effects of fragmentation on connectivity. This can aid stakeholders in making informed decisions for sustainable forest planning and management as well as assist them in ecological reconstruction efforts. This approach aims to reconnect forested areas to ensure optimal the continuity of habitats for endangered species.

## 2. Materials and Methods

This study examines the impact of deforestation on the compactness and connectivity of tree cover in the Carpathian Mountains. This analysis was conducted using succolarity, fractal fragmentation indices, and the local connected fractal dimension to analyse sets of generated and edited images within a case study framework. The images used were obtained from the Global Land Analysis and Discovery database [74] (https://glad.earthengine.app/view/global-forest-change; accessed on 19 April 2022). The following sections will describe each analytical method and type of image used. The goal is to provide a comprehensive understanding of the research approach.

### 2.1. Succolarity

This study investigates the use of succolarity, a metric which measures the degree of percolation of an image under analysis.

Succolarity ($\sigma$), along with fractal dimension (*FD*) and lacunarity ($\lambda$), are fractal measures evaluators. Fractal dimension indicates how much an object occupies its underlying metric space, while lacunarity examines the distribution of gaps or lacuna [75]. In contrast, succolarity quantifies the ability of "virtual water" to flow in an image. Succolarity is defined as an assessment of the degree to which filaments facilitate percolation or flow through an image [63].

To analyse forest patch fragmentation and connectivity from an ecological perspective, we have adopted a specific convention for our succolarity analysis. We interpret forest patches extracted from satellite images as interconnected or isolated lakes/seas, while non-forested or deforested surfaces are considered analogous to dry land between the lakes/seas.

The forest patches allow virtual water to infiltrate and flow, while the non-forested surfaces, including the deforested areas, act as obstacles around which virtual water must detour or come to a halt. Figure 1a,b depict the interconnection scenario among three hypothetical forest patches (similar to three lakes/seas). Virtual water can flow from one patch to another depending on the analysed flow direction. In this example, the analysis direction is from top to bottom (Figure 1b—green arrow). Figure 1c shows that, if the three forest patches are interconnected, virtual water from patch 1 reaches both patch 2 and patch 3 through connecting pathways, similar to water from the first lake feeding the second and then the third lake (with red is depicted percolation and with blue non-percolation after the obstacle).

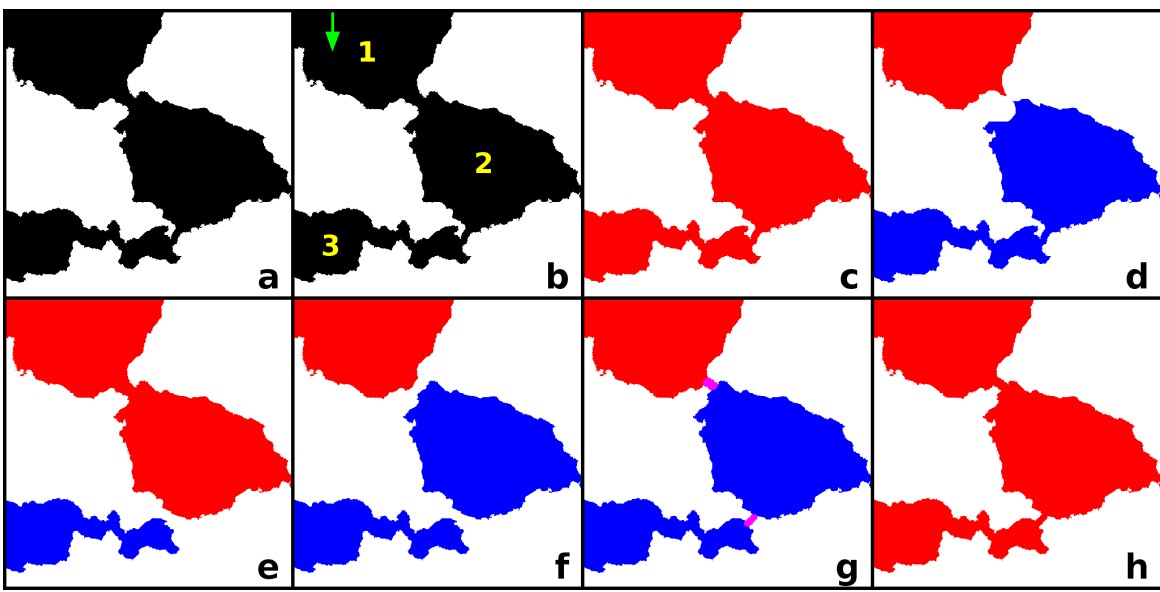

**Figure 1.** The interconnection scenario among three hypothetical forest patches, which are similar to three lakes. The figure demonstrates how succolarity quantifies the degree of fragmentation and connectivity of forest patches, the effects of the disappearance of connectivity, and the effects of creating corridors that connect forest patches (similar to the creation of channels connecting lakes): (**a**). Original image: (**b**). Identify the three connected forest patches (1, 2, and 3) and establish the analysis direction as top-to-bottom using the green arrow. (**c**). Highlight connectivity through percolation (indicated by the red colour) in all three patches as they are connected. (**d**). The effect of the disruption of connectivity between forest patches 1 and 2 means that percolation is allowed only in patch 1 (red colour) and is blocked in patches 2 and 3 (blue colour). (**e**). The effect of the disruption of connectivity between forest patches 2 and 3 leads to percolation being allowed only in patches 1 and 2 (red colour) and being blocked in patch 3 (blue colour). (**f**). The impact of disrupted connectivity between forest patches 1 and 2 and between patches 2 and 3 is demonstrated by percolation only being allowed in patch 1 (red colour) and blocked in patches 2 and 3 (blue colour). (**g**). To address this, connectivity corridors are created between unconnected patches, specifically between patch 1 and 2 and between patch 2 and 3. (**h**). As a result, connectivity highlighted by percolation is once again present in all three patches, as shown in (**c**), due to the establishment of connecting corridors.

If the interconnection between patches, similar to channels through which lakes communicate with each other, is disrupted (as shown in Figure 1d), only two out of the initially interconnected three patches remain connected but without percolation. This is because an obstacle has emerged, preventing the flow of virtual water. As a result, virtual water only flows within the first patch, leaving the second and third patches isolated. This prevents animals from moving between patches 2 and 3 to patch 1. If the connection between patch 2 and patch 3 is disrupted due to deforestation, patch 1 and 2 will still be connected, allowing for virtual percolation, while patch 3 will be isolated (refer to Figure 1e).

If deforestation eliminates both connections, all three patches become isolated, and percolation only occurs in the first patch when analysed from top to bottom (see Figure 1f). However, if new forest corridors are created through reforestation (see Figure 1g, magenta colour), the three patches become interconnected, restoring mobility between them (Figure 1h) (similar to the construction of channels which interconnect three isolated lakes).

Succolarity quantifies the degree of percolation in an image, representing the volume of liquid that can pass through. It measures how much of a given liquid can penetrate an image through black pixels, considering white pixels as obstacles when analysing 2D images.

The succolarity algorithm assesses the ability of virtual water to percolate in a given direction, providing insight into the spatial compaction and connectivity characteristics of

objects (such as tree cover). Each pixel is categorised as either empty (black pixels, symbolizing fluid access paths in the case study, i.e., tree cover) or filled with an impenetrable mass (white pixels, symbolizing obstacles in the case study, i.e., areas without forest, including deforestation effects) to assess this percolation capability.

De Melo and Conci (2013) describe the principles of calculating percolation ability using a box-counting approach for 2D images [65]. Thus, the virtual water pressure values applied to the images depend on the number of pixels and the scale of box-counting in each direction.

Fluid flow is simulated through all the connected tree cover pixels (black pixels) in each direction, while fluid is restricted from flowing through the obstacle pixels (white pixels). The area of fluid flow being targeted is divided into equal-sized boxes in the images, which are referred to as "*BS(k)*". Here, "*k*" represents the number of possible box-counting partitions in an image. The occupancy percentage for each image box size is directly measured on the images and noted as "*OP(BS(k))*". The pressure value for each box size k is calculated using Equation (1).

$$\sum_{k=1}^{n} OP(\text{BS}(k)) \text{xPR}(\text{BS}(k), \text{pc}) \tag{1}$$

where "*n*" represents the number of possible divisions, and "*PR(BS(k),pc*" represents the pressure at the centroid of box "*k*" in the scale of consideration.

To ensure that the succolarity value is dimensionless, this value is obtained by dividing the pressure occupancy value by the maximum possible pressure occupancy value (Equation (2)).

$$\sigma(\text{BS}(k), \text{dir}) = \frac{\sum_{k=1}^{n} OP(\text{BS}(k)) \text{xPR}(\text{BS}(k), \text{pc})}{\sum_{k=1}^{n} OP(\text{BS}(k)) \text{xmaxPR}(\text{BS}(k), \text{pc})} \tag{2}$$

The variable "*dir*" represents the direction, such as left-to-right (*l2r*) or right-to-left (*r2l*) and top-to-down (*t2d*) or down-to-top (*d2t*). The variable "*max*" represents the maximum pressure assignment.

The succolarity value, which ranges from 0 to 1, reflects the permeability of the image. If an image contains both white obstacle pixels and black percolation pixels, the succolarity values will fall between 0 and 1. A succolarity of 0 occurs when the entire image is white (waterproof image) or when the border (edge) in the flow direction is completely white, preventing water penetration. A value of 1 indicates unrestricted fluid flow, where the inner image is black. In this case study, succolarity is observed to increase with a higher permeability, which is characterized by larger interconnected black patches (tree cover) amid the white background (not forested or deforested areas). On the other hand, a lower permeability, which leads to fewer or smaller black connected patches, results in lower succolarity values.

To capture variations in the percolation direction, there can be calculated five succolarity values based on the direction of the virtual water flow: top-to-down (*t2d*), down-to-top (*d2t*), left-to-right (*l2r*), right-to-left (*r2l*), and the mean succolarity for all four directions. The mean succolarity provides a comprehensive overview of the percolation characteristics across different flow directions.

### 2.2. Potential Succolarity

Potential succolarity is a new concept that is proposed in this paper to indicate the extent to which the degree of percolation could be extended if obstacles isolating part of the patches could be penetrated.

The potential succolarity is calculated using Equation (3):

$$Potential\ succolarity = \frac{black\ pixels}{total\ pixels} \tag{3}$$

Potential succolarity does not distinguish between the four directions but rather provides a global overview.

### 2.3. Delta (Δ) Succolarity

Δ succolarity is also proposed as a new concept, representing the difference between potential succolarity and succolarity (Equation (4)):

$$\Delta\ succolarity = Potential\ succolarity - succolarity \tag{4}$$

Δ succolarity provides information on how much percolation could be further extended, considering the four flow directions, if obstacles isolating part of the patches could be penetrated.

Succolarity, potential succolarity, and Δ succolarity were calculated using the succolarity plugin from the ComsystanJ (Complex Systems Analysis for ImageJ) [76] collection plugins version 1.1.2 of Fiji/ImageJ2 2.14.0/1.54h java 1.8.0_322 64-bit [77]. Succolarity is a binary algorithm that operates on a [0, >0] scale by flooding the black pixels of a binary image. Therefore, the input images must be in an 8-bit binary format. The number of boxes for the regression parameters was set to 7, 8, 9, 10, and 11 for image scales of $64 \times 64$, $128 \times 128$, $256 \times 256$, $512 \times 512$, and $1024 \times 1024$, respectively. For the Romanian Carpathian forest area images, which had an image scale of $1000 \times 1000$ pixels, the number of boxes for the regression parameters was set to 10. The scanning type selected was the raster box.

### 2.4. Fractal Fragmentation Index (FFI)

The fractal fragmentation index (*FFI*) is a fractal index that uses multiscale fractal techniques to measure the degree of fragmentation or compression exhibited by objects occupying a space [54]. The *FFI* can also assess the deviation of the shape of each object from a geometric Euclidean shape. The calculation of the *FFI* is determined using the following Equation (5):

$$FFI = D_A - D_P = \lim_{\varepsilon \to 0} \left( \frac{\log N(\varepsilon)}{\log \frac{1}{\varepsilon}} \right) - \lim_{\varepsilon \to 0} \left( \frac{\log N'(\varepsilon)}{\log \frac{1}{\varepsilon}} \right) \tag{5}$$

where *FFI* is the fractal fragmentation index; $D_A$ is the fractal size of the summed areas, representing the size of the box required to cover the object area; $D_P$ is the fractal size of the accumulated perimeters, representing the size of the box required to cover only the object perimeter; $\varepsilon$ represents the size of the box used in the analysis; $logN(\varepsilon)$ represents the number of contiguous and non-overlapping boxes required to cover the object area; and $logN'(\varepsilon)$ represents the number of contiguous and non-overlapping boxes required to cover only the object perimeter.

There are three situations, as follows:

- *FFI* = 0: This situation occurs when forest areas are very small, typically 1–4 pixels for an object. In such cases, the perimeter image using the outline function appears to be similar to the area image, with $D_A = D_P$. In practice, when analysing very small fractal objects or entities, it is important to recognise the limitations associated with the accurate extraction of their contours. The accuracy of contour extraction can be compromised when dealing with diminutive fractal entities, potentially affecting the reliability and interpretability of the fractal fragmentation index (*FFI*) calculations for these specific entities. Careful consideration of these limitations is essential to ensure the meaningful interpretation of *FFI* results, particularly in cases involving small fractal structures.
- 0 < *FFI* < 1: In scenarios where the areas occupied by objects are large and compact, the *FFI* value is high and close to 1, indicating that the objects are densely packed. Conversely, if the objects are more scattered, smaller, or fewer in number, or if they have a tentacular, irregular, and sprawling pattern, the *FFI* value decreases and approaches 0.

- *FFI* = 1: This situation is recorded when the area is geometrically perfect and 100% compact, without any discontinuity ($D_A$ = 2 and $D_P$ = 1) [54].

### 2.5. Fractal Fragmentation and Disorder Index (FFDI)

The fractal fragmentation and disorder index (*FFDI*) uses multiscale fractal techniques to quantify the degree of fragmentation, compaction, and spatial disorder exhibited by the objects filling a space [39]. This index is an extension of the *FFI* and incorporates aspects of the information dimension ($D_1$) [78]. Similar to the *FFI*, the *FFDI* distinguishes patterns of spatial organisation in processed images.

The calculation of the *FFDI* is determined using the following Equation (6):

$$\text{FFDI} = D_1(1 - \text{FFI}) = \left(\lim_{\varepsilon \to 0} \sum_{i=1}^{N(\varepsilon)} \frac{m_i(\varepsilon)\log(m_i(\varepsilon))}{\log(\varepsilon)}\right)\left(1 - \left(\lim_{\varepsilon \to 0}\left(\frac{\log N(\varepsilon)}{\log\frac{1}{\varepsilon}}\right) - \lim_{\varepsilon \to 0}\left(\frac{\log N'(\varepsilon)}{\log\frac{1}{\varepsilon}}\right)\right)\right) \quad (6)$$

where $mi = Mi/M$, $M_i$ is the number of points in the *i*th box; $M$ is the total number of points of the objects; $\varepsilon$ represents the size of the box used in the analysis; $logN(\varepsilon)$ represents the number of contiguous and non-overlapping boxes required to cover the object area; and $logN'(\varepsilon)$ represents the number of contiguous and non-overlapping boxes required to cover only the object perimeter.

In this Equation (6), $1 - FFI$ is used because *FFI* = 0 indicates fragmentation, and *FFI* = 1 indicates a lack of fragmentation. The *FFDI* ranges from 0 to 2. The maximum value of the *FFDI* approaches 2 when the objects are highly disordered and fragmented, indicating a highly disordered spatial arrangement. Conversely, the lowest value approaches 0 when the objects are weakly disordered and compact, indicating a more ordered and compact spatial organisation.

The *FFI* and *FFDI* were calculated using the fractal fragmentation indices plugin available in the ComsystanJ collection plugins [76] for Fiji/ImageJ2 2.14.0/1.54h [77]. These plugins provide binary algorithms that operate on a [0, >0] scale by analysing the white pixels of a binary image. Therefore, it is imperative that the input images are in an 8-bit binary format, inversely configured compared to the images used for the succolarity analysis.

To be consistent with the succolarity analysis, the number of boxes for the regression parameters was set to 7, 8, 9, 10, and 11 for image scales of 64 × 64, 128 × 128, 256 × 256, 512 × 512, and 1024 × 1024, respectively. For the Romanian Carpathian forest area images, which had an image scale of 1000 × 1000 pixels, the number of boxes for the regression parameters was set to 10.

Note that succolarity-compatible images require inversion using Fiji/ImageJ2 2.14.0/1.54h (Edit–Invert). This inversion ensures that white pixels correspond to the tree cover, while black pixels represent the background areas without forest, including deforestation effects.

### 2.6. Local Connected Fractal Dimension (LCFD)

The local connected fractal dimension (*LCFD*) is a fractal index used to quantify local connectivity. It is expressed according to Equations (7) and (8):

$$M(\epsilon) \propto F\epsilon^{LCFD} \quad (7)$$

$$LCFD = \frac{log[M(\epsilon)]}{log(\epsilon)} \quad (8)$$

where $F$ is a mass prefactor, and $M(\epsilon)$ is the number of locally connected pixels (an eight-neighbour connection) in a *box $\varepsilon$* [53,79].

The *LCFD* value is 1 if the object is a straight one-dimensional line and 2 if the object is two-dimensional and completely covered. The determination of connectivity is based on considering pixels within the 8 × 8 environment of the starter pixel as connected, following the basic rule for identifying connected sets within a given arbitrary distance around the base pixel.

*LCFD* estimation was performed using the FracLac plugin [80] for the Fiji/ImageJ2 2.14.0/1.54h [77]. In this analysis, binary images were imported into FracLac as a binary stack. The local connected fractal dimension option was selected with black as the background for the fractal connectivity analysis. The result included an average of all connectivity values per image.

### 2.7. Image Generation for Testing and Validating Succolarity

The aim of generating images was to evaluate and validate succolarity's ability to quantify the degree of fragmentation, compaction, and interconnection. This evaluation aimed to assess succolarity's performance in relation to directionality.

Firstly, five sets, each containing six images, were generated using the Fiji/ImageJ2 2.14.0/1.54h software [77]. The sets consisted of three isotropic images (black square, cross, and white square) as well as three anisotropic images (three-quarter square, half square, and quarter square). The images were generated at scales of 64 × 64, 128 × 128, 256 × 256, 512 × 512, and 1024 × 1024 pixels (Figure 2). This process aimed to assess the sensitivity of succolarity to variations in image scale, mirroring the approach taken in our fractal analysis. To prevent potential biases introduced by automatic resizing and other error sources, it was essential to ensure uniformity in scale (pixel size) and extent when comparing images for the fractal analysis. As highlighted by Loke and Chisholm (2022), this standardization applies to both algorithmically generated images and those derived from empirical data [81].

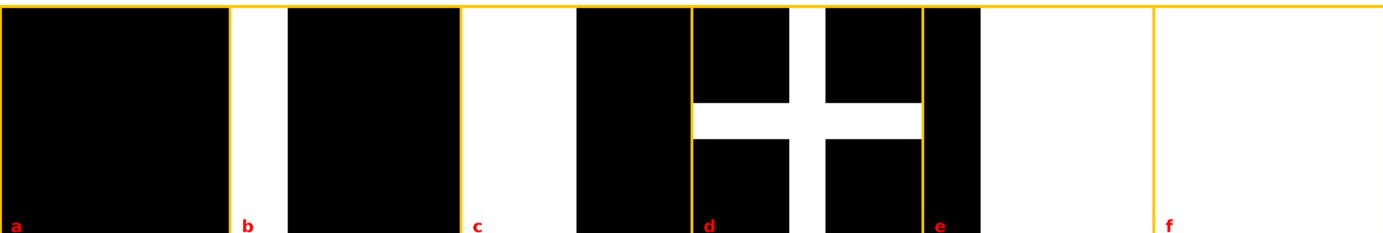

**Figure 2.** Images generated for succolarity testing and validation: (**a**). black square; (**b**). three-quarter square; (**c**). half square; (**d**). cross; (**e**). quarter square; and (**f**). white square. The images were generated at scales of 64 × 64, 128 × 128, 256 × 256, 512 × 512, and 1024 × 1024 pixels.

Secondly, the Image generator—Fractal HRM from ComsystanJ collection plugin [76] was used within the Fiji/ImageJ2 2.14.0/1.54h software to generate a hierarchically structured random map (HRM) image. The HRM image was created using a recursive algorithm inspired by the curdling and random trema generation method introduced by Plotnick [27,63]. HRMs are maps of pixels randomly assigned values of 0 or 1, with varying probability distributions in different regions. These maps can simulate landscapes with varying levels of clustering, fragmentation, and connectivity, achieved through the processes of curdling and random trema. It is important to note that two surfaces may share the same overall fill space and yet diverge in their degree of curdling, which denotes the clustering or nucleation of the filled space. The unfilled space is created through a random process called trema, which involves placing holes with a defined shape on the mathematical surface [25,26,63].

The HRM image for this experiment was generated at a scale of 512 × 512 pixels, corresponding to 262,144 pixels, with a fixed number of 512 rows and columns (*M*) (Figure 3a). The image falls under category s1, which is denoted as "high" (refer to [39] for comprehensive details). Specific probability values were assigned: $p1 = 0.5$ and $p2 = p3 = 1$. The variables $p1$, $p2$, and $p3$ represent the percentage of space occupied by HRM objects at different grain sizes, expressed as a fraction (e.g., 0.5 corresponds to 50%). Specifically, p1 corresponds to a grain size of 50 × 50 pixels, p2 to 5 × 5 pixels, and p3 corresponds to 1 × 1 pixel. These variations in grain size enable the examination of HRM objects at different scales within the fixed scale of the 512 × 512-pixel HRM image. The image produced is

anisotropic, making it appropriate for a succolarity analysis in all four directions: *t2d*, *d2t*, *l2r*, and *r2l* (Figure 3b–e).

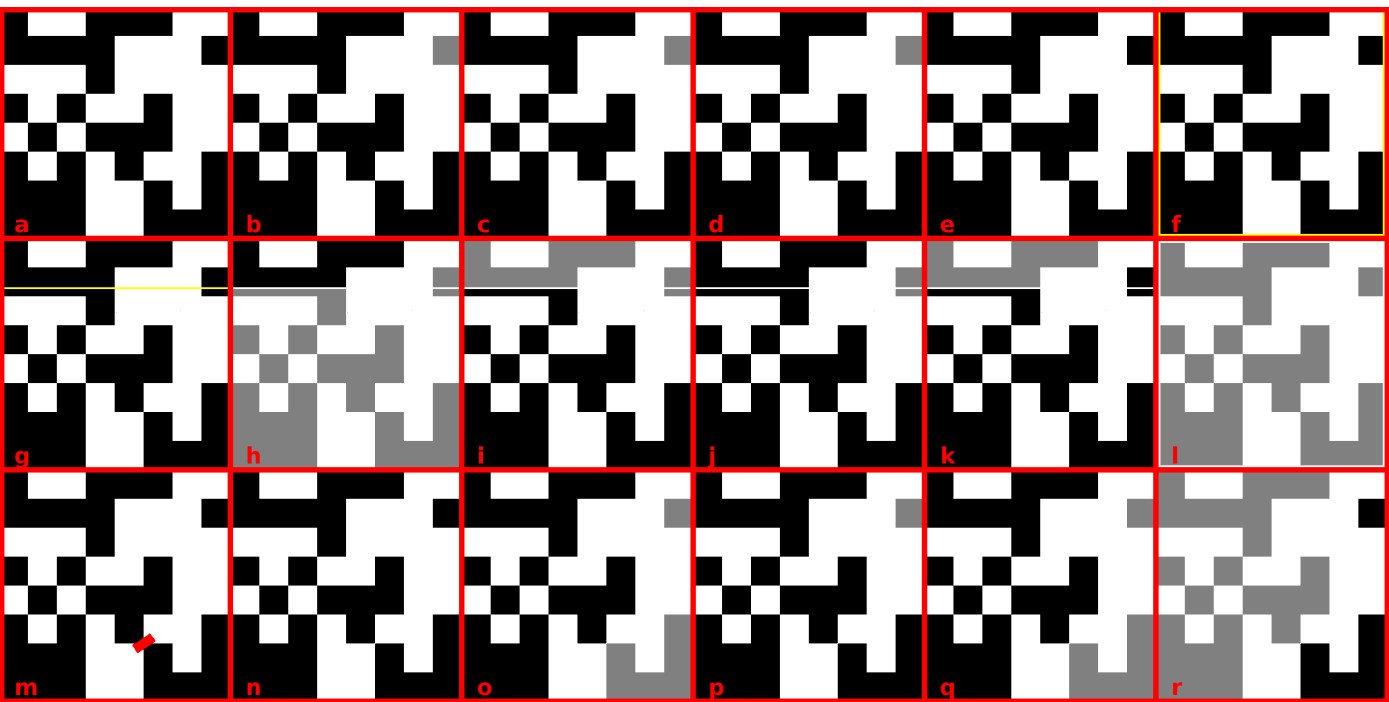

**Figure 3.** Hierarchically structured random map (HRM) images generated and edited for succolarity testing and validation: (**a–e**). original image; (**a**). the HRM image is generated using ComsystanJ, with assigned probability values of p1 = 0.5 and p2 = p3 = 1; (**b–e**). visualisation of percolation in the four directions of analysis ((**b**). top-to-down; (**c**). down-to-top; (**d**). left-to-right; and (**e**). right-to-left); (**f**). encircled by an unbroken barrier (yellow border); (**g**). a barrier line has been added (yellow); (**h–k**). visualisation of percolation in the four directions of analysis for an HRM image with a barrier line added ((**h**). top-to-down; (**i**). down-to-top; (**j**). left-to-right; and (**k**). right-to-left); (**l**). visualisation of percolation for HRM image encircled by an unbroken barrier; (**m**). HRM image with two pixels removed in a connection area (the red line indicates the area of the removed connectivity); (**n**). HRM image with two pixels removed in a connection area; and (**o–r**). visualisation of percolation on the four directions of analysis for an HRM image with two pixels removed in a connection area ((**o**). top-to-down; (**p**). down-to-top; (**q**). left-to-right; and (**r**). right-to-left). The images display black pixels that enable percolation, allowing virtual water to flow. White pixels represent obstacles, while grey pixels are located beyond the obstacles and are unreachable by virtual water (they are not included in the quantification of succolarity but indicate the potential for percolation if water penetrated the obstacle).

Thirdly, the HRM image was edited to enable more detailed tests of succolarity.

- The first edit included adding a horizontal white line, four pixels thick, in the initial optimum of the image (Figure 3g; for better visualisation, the line is shown in yellow in the figure). When a barrier is completely impenetrable, fluid flow stops, and the black pixels beyond the barrier are not measured. This is because the barrier acts as an impermeable layer, blocking the virtual liquid above. A new concept, Δ succolarity, was introduced to quantify these black pixels located beyond the barrier. Together with the quantifiable pixels located in front of the barrier, they formed the potential succolarity, another proposed concept. The aim of this approach was to measure the effect of a continuous obstacle on the flow of virtual water. The evaluation focused on the four directions of analysis (*t2d*, *d2t*, *l2r*, and *r2l*) (Figure 3h–k), with a particular emphasis on *t2d* and *d2t* (Figure 3h,i). Figure 3h–k depict the four percolations limited by the presence of the barrier, corresponding to the four analysed directions. To aid

visualization, the black pixels located beyond the barrier and thus not quantified are represented in grey.

- In the second edit, the HRM image was framed by a white border that was 12 pixels wide (Figure 3f). The purpose of this edit was to measure succolarity when the image was completely impermeable to water (Figure 3l). In the presence of a white border, which acts as a continuous external obstacle in all four directions, all black pixels that could potentially allow percolation are considered "inactive". This is because the border prevents virtual liquid from penetrating the image. In this scenario, these "inactive" black pixels represent both the potential and Δ succolarity, as their succolarity is equal to 0. These "inactive" black pixels are located beyond the image border (see Figure 3f) and are displayed in grey for clarity (see Figure 3l).

- In the third edit, two black pixels that connected two patches in the original image were removed (as shown by the red line in Figure 3m and by the white in Figure 3n). Figure 3o–r show the four percolations corresponding to the four analysed directions. Two black pixels were removed, creating a small white barrier between the two patches. It is evident that breaking a connection between two patches deactivates some black pixels beyond the rupture of that connection. These pixels no longer facilitate "flow" and become "inactive" due to the obstacle created. To improve visualization, grey pixels located beyond the connection rupture and, therefore, not quantified represent these black pixels. The aim was to identify changes in succolarity when interconnections became obstacles (Figure 3o–r).

These second and third edits provided essential support for validating the two new concepts proposed in this study: potential succolarity and Δ succolarity.

### 2.8. Tree Cover Images of the Romanian Carpathians Mountains

In the current research, the tree cover database for 2001–2021 from a previous research study (available as supplementary material [39] was used (https://static-content.springer.com/esm/art:10.1007/s10980-023-01640-y/MediaObjects/10980_2023_1640_MOESM4_ESM.zip; accesed on 19 September 2023). The images were processed from the Global Land Analysis and Discovery database [74] (https://glad.earthengine.app/view/global-forest-change; accessed on 19 April 2022). The methodology for obtaining these images and preprocessing them to generate the final images is described in [39]. The study area is a significant geographical region with mountainous terrain forming a circular arc that stretches from the northern border with Ukraine to the southern border with Serbia (Figure 4).

Ten stacks of twenty-two images each were analysed, one image for each year of the analysed time interval, from 2000 to 2021. The ten stacks corresponded to the ten mountain groups of the Romanian Carpathians: the Northern Eastern Carpathians, Central Eastern Carpathians, Southern Eastern Carpathians, Bucegi, Făgăraș, Parâng, Retezat–Godeanu, Banat, Poiana Ruscă, and Apuseni mountain groups (Figure 5). The size of all the images was 1000 × 1000 pixels. Black pixels corresponded to forest cover and white pixels to non-forest or deforested areas for the succolarity analysis and the opposite for the *FFI*, *FFDI*, and *LCFD* analyses.

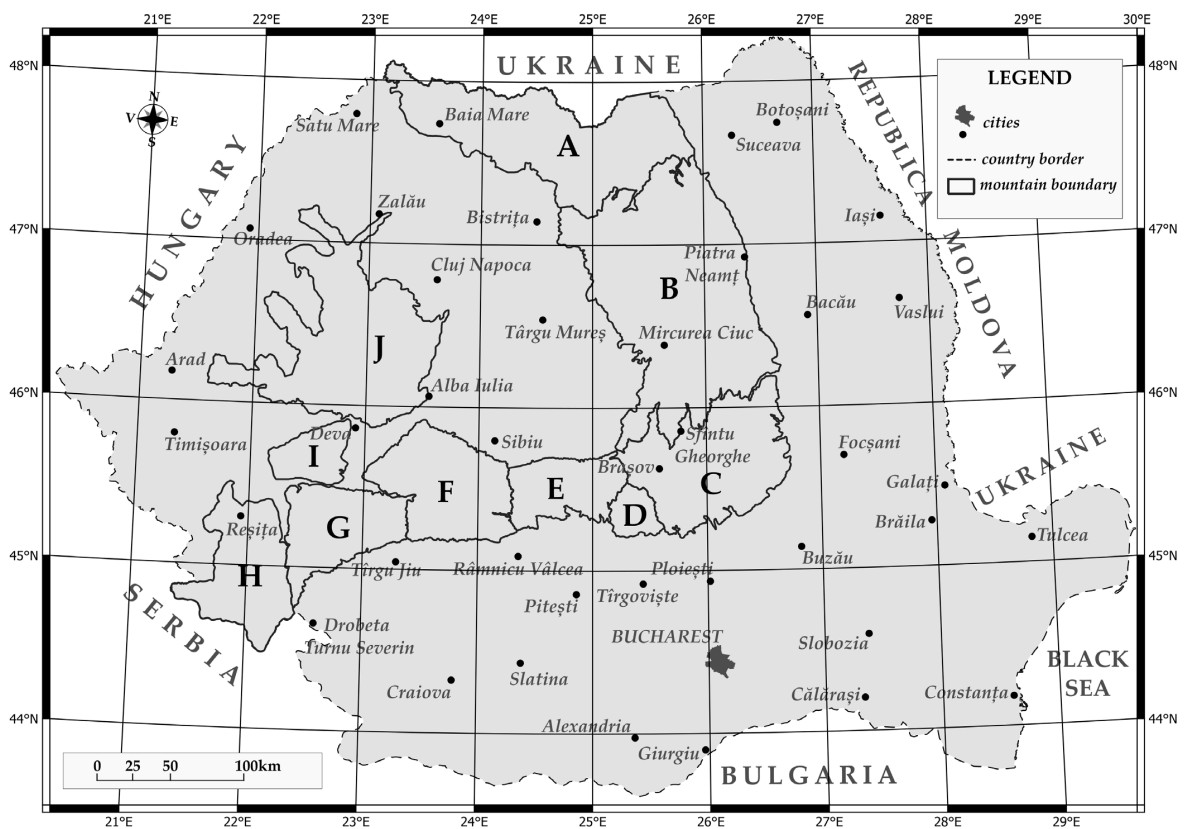

**Figure 4.** The geographic location of the Romanian Carpathians mountains: (**A**) northern group of Eastern Carpathians; (**B**) central group of Eastern Carpathians; (**C**) southern group of Eastern Carpathians; (**D**) Bucegi Mountains group; (**E**) Făgăraș Mountains group; (**F**) Parâng Mountains group; (**G**) Retezat–Godeanu Mountains group; (**H**) Banat Mountains; (**I**) Poiana Ruscă Mountains; and (**J**) Apuseni Mountains.

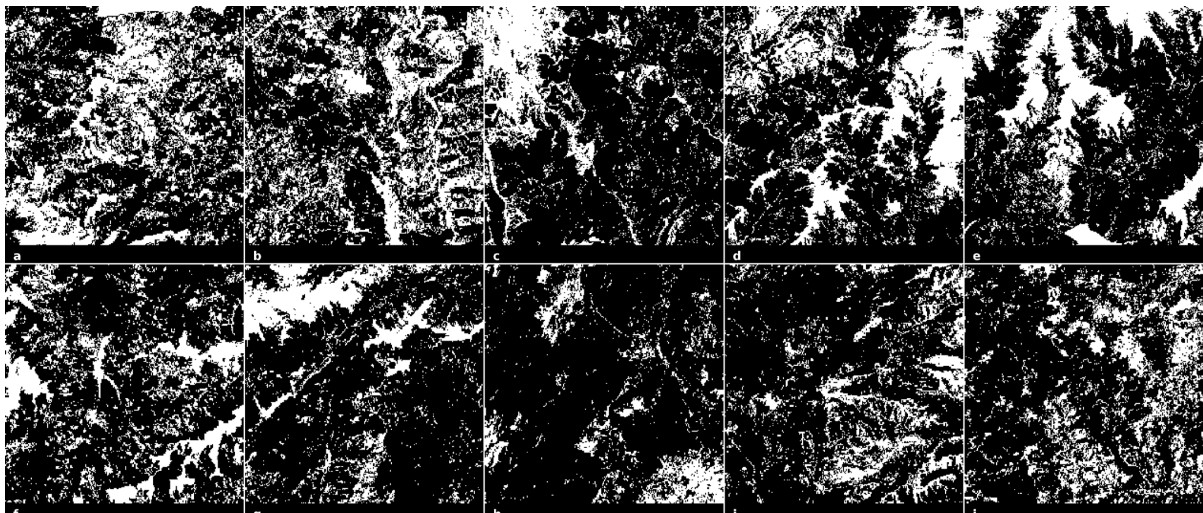

**Figure 5.** Examples of binary images of tree cover from 2021: (**a**) northern group of Eastern Carpathians; (**b**) central group of Eastern Carpathians; (**c**) southern group of Eastern Carpathians; (**d**) Bucegi Mountains group; (**e**) Făgăraș Mountains group; (**f**) Parâng Mountains group; (**g**) Retezat–Godeanu Mountains group; (**h**) Banat Mountains; (**i**) Poiana Ruscă Mountains; and (**j**) Apuseni Mountains. The dimension of all the images is 1000 × 1000 pixels. White pixels correspond to forest cover and black pixels correspond to non-forest cover. The accompanying images are 1000 × 1000 pixels and use black pixels to represent forest cover and white pixels to represent non-forest cover.

## 3. Results

This section presents the results of the succolarity analysis, focusing on the impact of the black patches' sizes and their connectivity on the obtained values. Succolarity values, which range from 0 to 1, serve as a significant indicator for assessing image permeability. The importance of the interaction between white and black pixels is highlighted by the distinction between low and high succolarity. The variability in succolarity is revealed by exploring the dimensions and connections of black patches, providing a detailed perspective on how these features impact image permeability.

The study used test images from Figures 1 and 2 to explore the capability of succolarity in quantifying the degree of fragmentation/compactness and connectivity of areas with black pixels that facilitate percolation against a background composed of white obstacle pixels. Additionally, the test images were analysed for comparison with three fractal indices: *FFI*, *FFDI*, and *LCFD*. After validation, succolarity was used to investigate the impact of deforestation on the density and connectivity of forest fragments over a 22-year period in ten mountainous regions of the Romanian Carpathians.

### 3.1. Invariance of Succolarity to Image Scale

This section explores the consistent behaviour of succolarity with respect to image scale. The analysis reveals that succolarity remains invariant across varying image scales, consistently presenting identical values irrespective of an image's dimensions, whether these are $64 \times 64$, $128 \times 128$, $256 \times 256$, $512 \times 512$, or $1024 \times 1024$ pixels (refer to Figure 6a).

However, the three analysed fractal dimensions (*FFI* (Figure 6b), *FFDI* (Figure 6c), and *LCDF* (Figure 6d)) are sensitive to image size. The *FFI* values increase proportionally with image scale (ranging from $64 \times 64$ pixels to $1024 \times 1024$ pixels), which introduces a bias in the analysis. For example, the *FFI* for a half-square image with dimensions of $1024 \times 1024$ pixels is higher than the *FFI* for a quarter-square image with dimensions of $256 \times 256$ pixels. As a result, larger images, such as those with dimensions of $1024 \times 1024$ pixels, exhibit higher compactness values compared to smaller images, such as those with dimensions of $512 \times 512$ pixels, even if they have identical texture and structure. This difference is due to the larger number of pixels being analysed.

The *FFDI* values decrease as image dimensions increase, resulting in larger images ($1024 \times 1024$) showing higher compactness and order values while maintaining uniform texture and structure. The *LCDF* values show a slight increase, from smaller image dimensions ($64 \times 64$) to larger ones ($1024 \times 1024$), indicating a slightly higher degree of connectivity while maintaining consistent texture and structure.

In the case of black images, which consist solely of foreground pixels, they achieve an *FFI* value of 1 and an *FFDI* value of 0, regardless of the image's size. Conversely, white images consisting solely of background pixels yield an *FFI* score of 0 due to the absence of quantifiable pixels. As a result, there is no connectivity (*LCDF* = 0), making it impossible to calculate the informational dimension, resulting in the *FFDI* consistently being *NaN*.

It is important to note that, for the analyses of the *FFI*, the *FFDI*, and the *LCDF*, the images underwent inversion, with white being transformed into black and vice versa. However, to maintain consistency and comparability with succolarity, the image nomenclature was retained, even if the black images were converted to white and the white images to black. ConsystanJ assumes a black background and a white foreground for fractal dimension analyses. Therefore, an analysis of a black image without inversion would result in *FFI* = 0, while, for a white image, *FFI* = 1, because ComsystanJ for fractal dimension analyses assumes a black background and a white foreground.

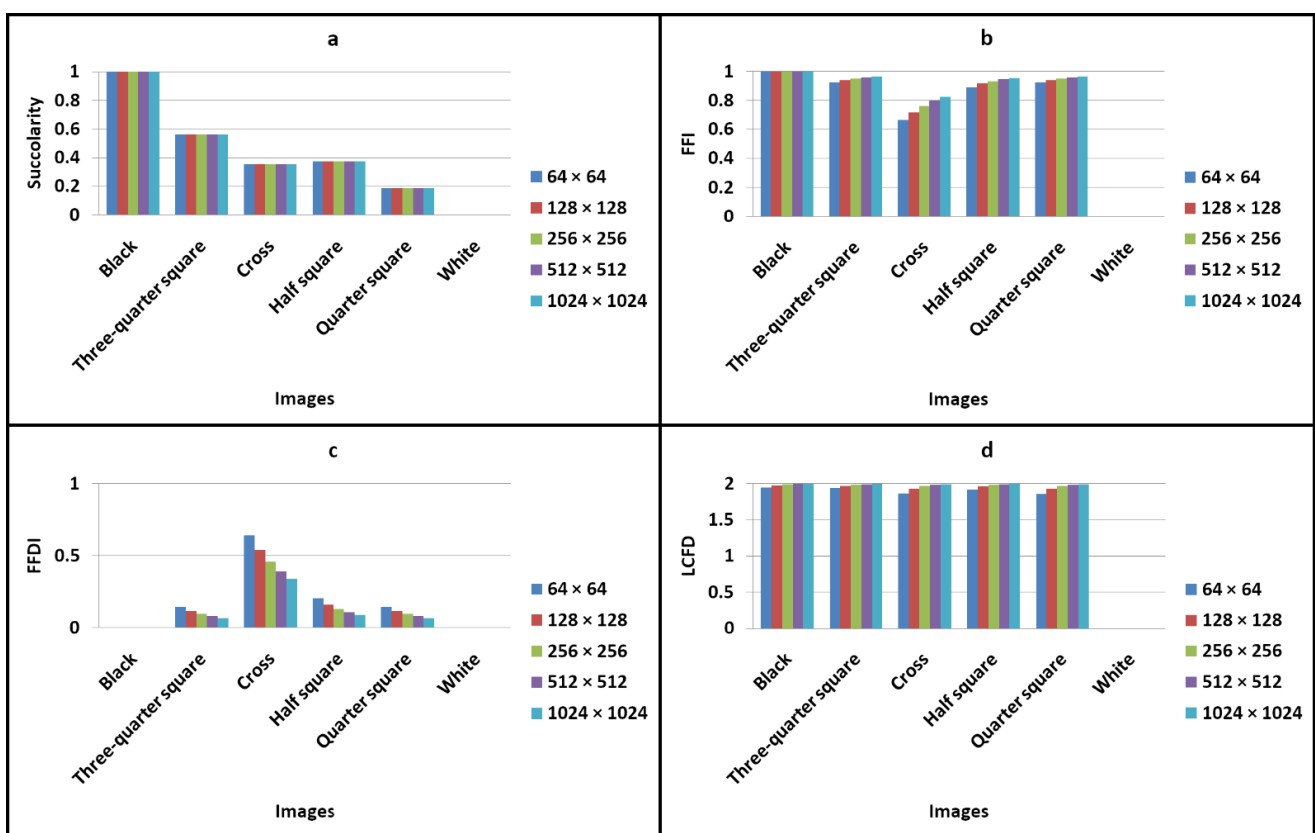

**Figure 6.** The influence of image scale on the analysis of succolarity, *FFI*, *FFDI*, and *LCFD*. The images analysed are those represented in Figure 1 and are analysed at scales of 64 × 64, 128 × 128, 256 × 256, 512 × 512, and 1024 × 1024 pixels. The results indicate that only succolarity is invariant to an image's scale (**a**), allowing for the easy comparison of results obtained from images of different scales. In contrast to succolarity (**a**), *FFI* (**b**) and *LCFD* (**d**) values increase proportionally with image scale, while *FFDI* (**c**) values decrease proportionally with image scale, making them sensitive to changes in image scale. It is important to note that for a black image, regardless of image scale, *FFI* has a maximum value and minimum *FFDI* (maximum compression), and for a white image, regardless of image scale, *FFI* has a maximum value and minimum *FFDI* (maximum fragmentation).

### 3.2. Examination of Succolarity and FFI, FFDI, and LCFD Metrics for the Generated Images

This section conducts a comprehensive analysis using succolarity, *FFI*, *FFDI*, and *LCFD* metrics on both the generated images and those that underwent subsequent modifications. The objective is to gain valuable insights into the permeability, fractality, and directional characteristics inherent in each image, providing a deep understanding of their structural properties. The assessments examine both the baseline features and the impact of specific alterations, such as removing two pixels, adding a white line, and introducing a white border. The findings reveal subtle variations in connectivity, permeability, and structural attributes, providing a nuanced perspective on the intricate behaviour of the images under different modifications.

Figure 7 shows the results of the succolarity, *FFI*, *FFDI*, and *LCFD* analyses for the six test images in Figure 2a–f. Figure 7a demonstrates that, for the isotropic images (black square, cross, and white square), succolarity remains constant in all four directions. The black square has the maximum succolarity value of 1, indicating the total permeability of virtual water through all the black pixels. In contrast, the white square has a succolarity of 0, indicating impermeability due to the absence of black pixels. The cross image has a succolarity of 0.36, indicating water penetration only in two symmetric spots out of four, regardless of the direction.

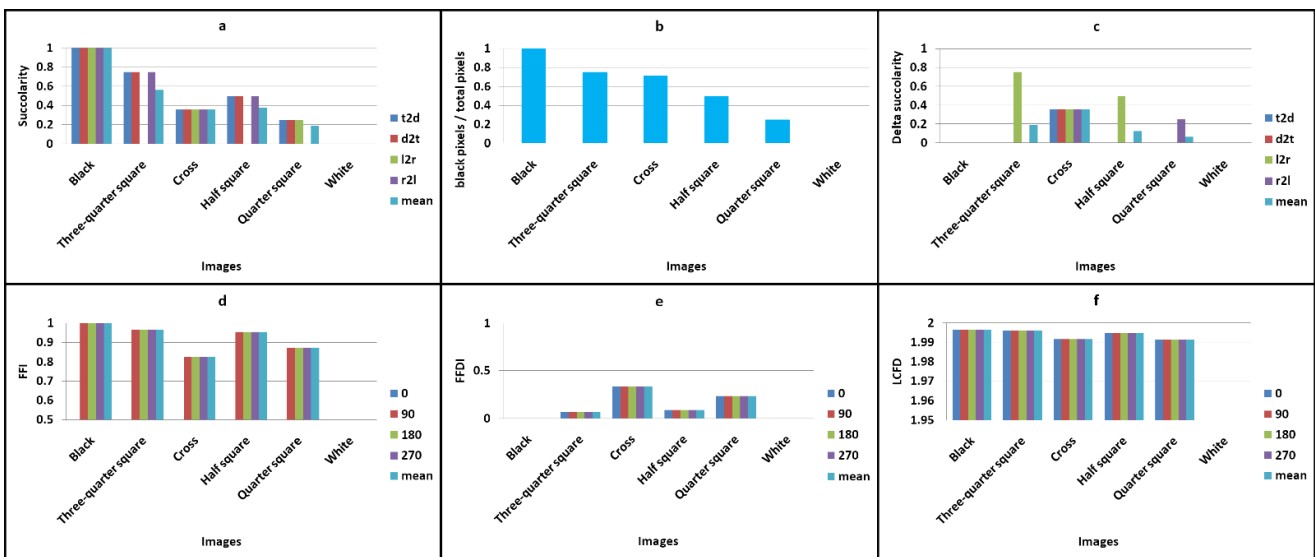

**Figure 7.** The results of the fractal analysis of the images in Figure 1. The analysis includes the following indices: (**a**). succolarity; (**b**). potential succolarity (a new index); (**c**). Δ succolarity (another new index); (**d**). *FFI*; (**e**). *FFDI*; and (**f**). *LCFD*. The scale of the images is 1024 × 1024 pixels. The results indicate that only succolarity can distinguish between isotropic images (black square, cross, and white square) and anisotropic images (three-quarter square, half square, and quarter square).

For the anisotropic images, such as the three-quarter square, the half square, and the quarter square, succolarity remains consistent in three directions. However, in the fourth direction, where an obstacle is encountered (represented by white), succolarity becomes 0. Therefore, the three-quarter square has a succolarity of 0.75 in three directions, with *l2r* = 0; the half square has a succolarity of 0.5 in three directions, with *l2r* = 0; and the quarter square has a succolarity of 0.5 in three directions, with *r2l* = 0. These results highlight the ability of succolarity to distinguish differences in permeability and structure between images, which is relevant for percolation and fractality analyses.

Figure 7b displays the ratio of black pixels that allow water percolation to the total number of pixels. This ratio represents the potential succolarity, as per Equation (3). The potential succolarity values for the generated images are as follows: black square = 1; three-quarter square = 0.75; cross = 0.71; half square = 0.5; quarter square = 0.25; and white square = 0.

Figure 7c shows the Δ succolarity, which is the difference between succolarity (Figure 7a) and potential succolarity (Figure 7b), as defined in Equation (4). The isotropic images for the black square (all the pixels allowing percolation; potential succolarity is equal to succolarity) and the white square (no black pixel potentially allowing percolation) have a value of 0 in all directions, as there are no additional obstacles or fragmentations. For the cross, the value is 0.36 if the white barrier can be penetrated without being eliminated. The Δ succolarity for the anisotropic images indicates a potential interconnection in the *l2r* direction, leading to an increase in succolarity by 0.75 (three-quarter square) and 0.5 (half square). In the *r2l*, there is a potential interconnection that could increase succolarity by 0.25 (quarter square).

Figure 7d shows the *FFI* values for the six images. Unlike succolarity, *FFI* does not differentiate based on the analysis direction for both the isotropic and anisotropic images. *FFI* quantifies the degree of fragmentation/compactness, including isolated unconnected patches, unlike succolarity. The five images exhibit perfect geometric objects (square, rectangle); therefore, the *FFI* values range between 0.83 (cross) and 1 (black square). The *FFI* of the white square is 0, as it does not have any foreground pixels.

The *FFDI* analysis (Figure 7e) shows a similar situation, but it is inversely proportional to *FFI*. The minimum value is observed for the black square (0, indicating no fragmentation

or disorder), and the maximum is 0.34 for the cross. The analysis of the white square cannot be conducted due to the absence of foreground pixels.

The analysis of *LCFD* (Figure 7f) produces similar results between the minimum value (1.991 for the quarter square) and the maximum value (1.997 for the black square). For the white square, *LCFD* is 0 due to a lack of connectivity. Therefore, *LCFD* has not proven to be useful in distinguishing the analysed images.

In Figure 8a, the HRM top image has a good connectivity between the black squares and exhibits relative isotropy with a succolarity of 0.5 in the *r2l* direction and 0.48 in the other three directions. The removal of two pixels (Figure 3h) results in a reduced connectivity in all four directions, with the most significant decrease observed in the *r2l* direction, from 0.5 to 0.11. The only direction that remains unchanged is *d2t*. Similarly, the introduction of a horizontal barrier in the first optimum (Figure 3h) leads to similar reductions in succolarity as the removal of two pixels, with the most significant decrease observed in the *t2d* direction, from 0.48 to 0.10. The direction that remains almost unchanged is *l2r*. When a white border is added around the image (Figure 3l), succolarity becomes 0 in all four directions, despite there being connectivity and fragmentation/compactness within the interior. This is because the virtual water cannot penetrate the image, reach the black pixels, and allow percolation.

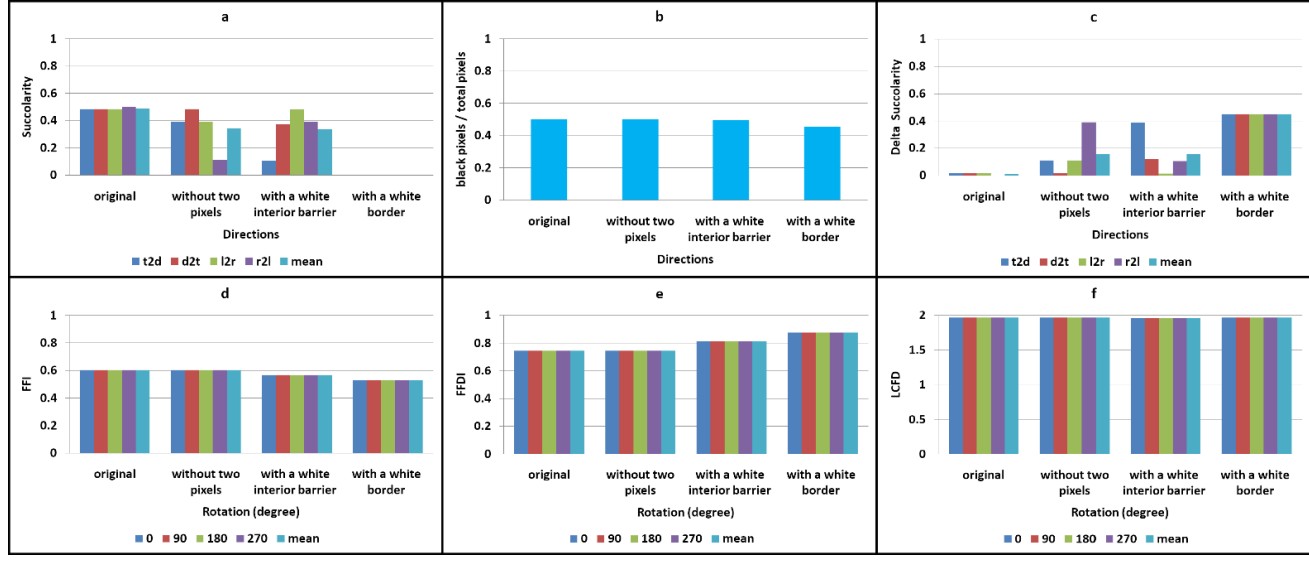

**Figure 8.** The results of the fractal analysis of the HRM images in Figure 3. The analysis includes the following indices: (**a**). succolarity; (**b**). potential succolarity (a new index); (**c**). Δ succolarity (another new index); (**d**). *FFI*; (**e**). *FFDI*; and (**f**). *LCFD*. The scale of the images is 512 × 512 pixels. The results show that only succolarity can be used to determine directionality, as the fractal dimensions analysed have been found to be insensitive to the direction of analysis.

The impact of removing two pixels, adding a white line, or including a white border on the ratio of black pixels to total pixels is minimal, as shown in Figure 8b. Figure 8c displays the results of the Δ succolarity analysis, which reflects the effects of these three modifications. However, for the image without two pixels and with a white interior barrier, succolarity is below 0.2 (almost 0.39 for images without two pixels in *r2l* and with a white interior barrier in *t2d*). In the original image, the Δ succolarity is approximately 0.01. In the case of the image with a white border, the Δ succolarity is 0.45. If the border were to be penetrated in one or more directions, the succolarity would increase from 0 to approximately 0.5. This implies a good potential for flow but requires the penetration of the border. The impact of the three adjustments on the *FFI*, the *FFDI*, and the *LCFD* for the barrier and border is negligible, with little variation observed when two pixels are removed (see Figure 8d–f). Figure 9a,c clearly illustrate the differences in directionality for the succolarity of the two image sets, while Figure 9b,d depict the Δ succolarity.

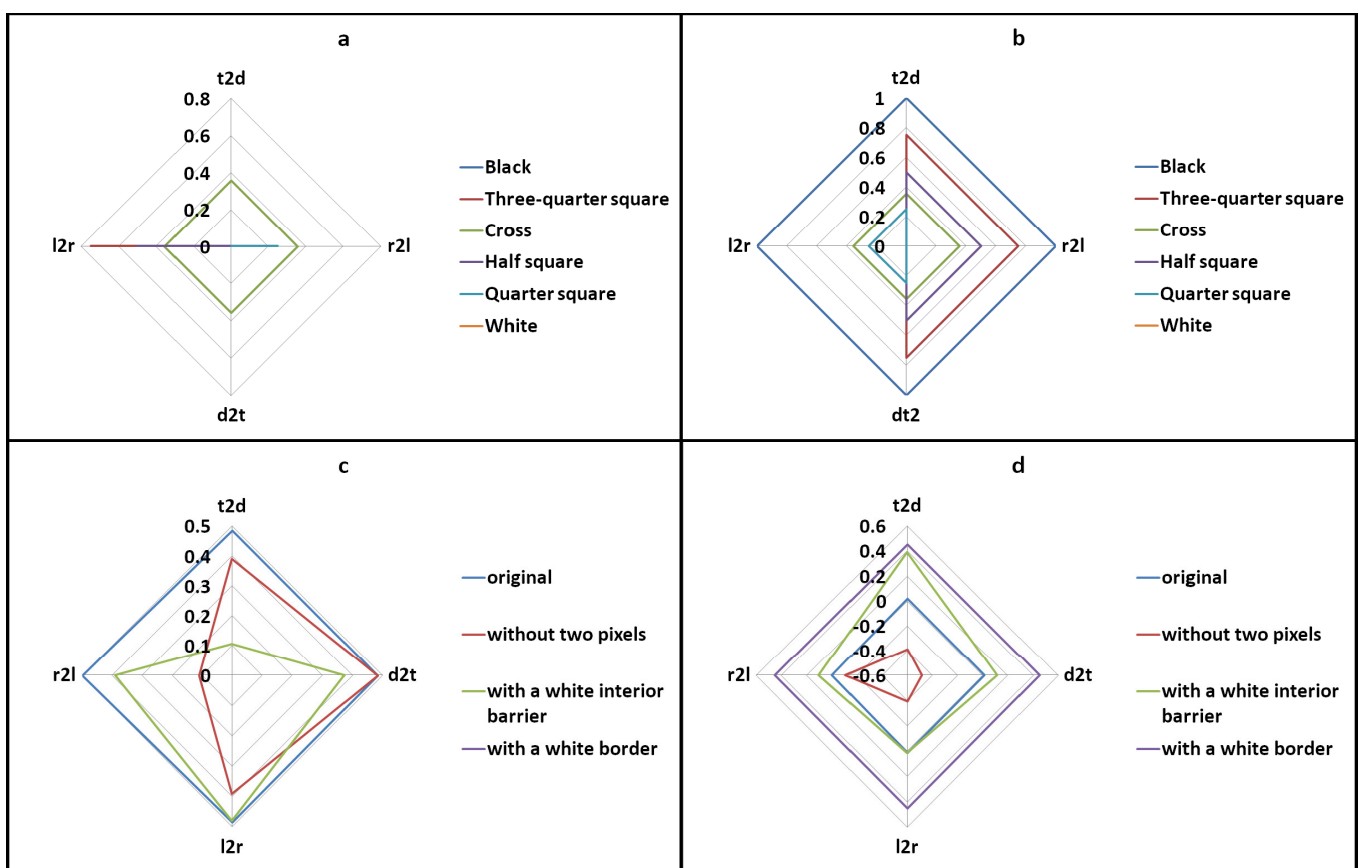

**Figure 9.** Analysis of succolarity and Δ succolarity according to the four directions of analysis (top-to-down, down-to-top, left-to-right, and right-to-left): (**a**). succolarity for the images in Figure 1; (**b**). Δ succolarity for the images in Figure 1; (**c**). succolarity for the images in Figure 3; and (**d**). Δ succolarity for the images in Figure 3.

*3.3. A Comprehensive Analysis of Forest Succolarity, Connectivity, and Fractality in the Romanian Carpathians*

This section provides a detailed analysis of the dynamics of succolarity, connectivity, and fractality in the Romanian Carpathians. The aim is to examine the structural changes in the forested landscape, with a focus on variations in the compactness and connectivity of forest patches, particularly in response to deforestation and natural losses.

Figures 10 and 11 display the local connectivity map for two mountain groups with opposing patterns. The northern group of the Eastern Carpathians is characterized by intense deforestation, which has caused both fragmentation and reduced connectivity between forest patches (Figure 10). In contrast, the Retezat–Godeanu group has experienced lower levels of deforestation, resulting in only minor fragmentation and reduced connectivity between forest patches (Figure 11).

Succolarity is an important metric as it indicates the level of fragmentation. Lower values suggest the presence of isolated and unconnected patches resulting from forest fragmentation.

Figure 12 shows the dynamics of tree cover across the 10 mountain groups, displaying the mean succolarity (Figure 12a) and the differences between the years 2000 and 2021 (Figure 12b). Four distinct patterns were identified:

- The Western Carpathians (the Banat, Poiana Ruscă, and Apuseni mountain groups) exhibited high compactness and connectivity with minimal deforestation, avoiding significant fragmentation or isolation of forest patches.
- The Parang and northern mountain groups displayed moderate compactness and connectivity with high dynamicity. Deforestation in Parang resulted in the creation of

isolated forest patches, while the northern group experienced extensive deforestation, leading to both continuous fragmentation and reduced connectivity. The Bucegi and Retezat mountain groups were identified as having moderate compactness and connectivity, with reduced dynamicity. Despite being the second-most impacted group by deforestation, the central group had low connectivity and compactness before 2001.

- Low compactness and connectivity (reduced dynamicity) were observed in the Fagaras and southern mountain groups, with minimal deforestation affecting only a few connectivity areas.

The differences in succolarity between 2000 and 2021 were most significant in the Northern Parang and central groups, indicating substantial fragmentation and connectivity loss (mean succolarity reduced by over 0.09). Moderate differences were observed in the Bucegi, Poiana Rusca, and Apuseni groups, with mean succolarity reductions ranging between 0.02 and 0.05. The Banat, southern, and Retezat groups exhibited the smallest differences, with mean succolarity reductions below 0.02. This suggests that deforestation has had a lesser impact on connectivity and fragmentation in these regions.

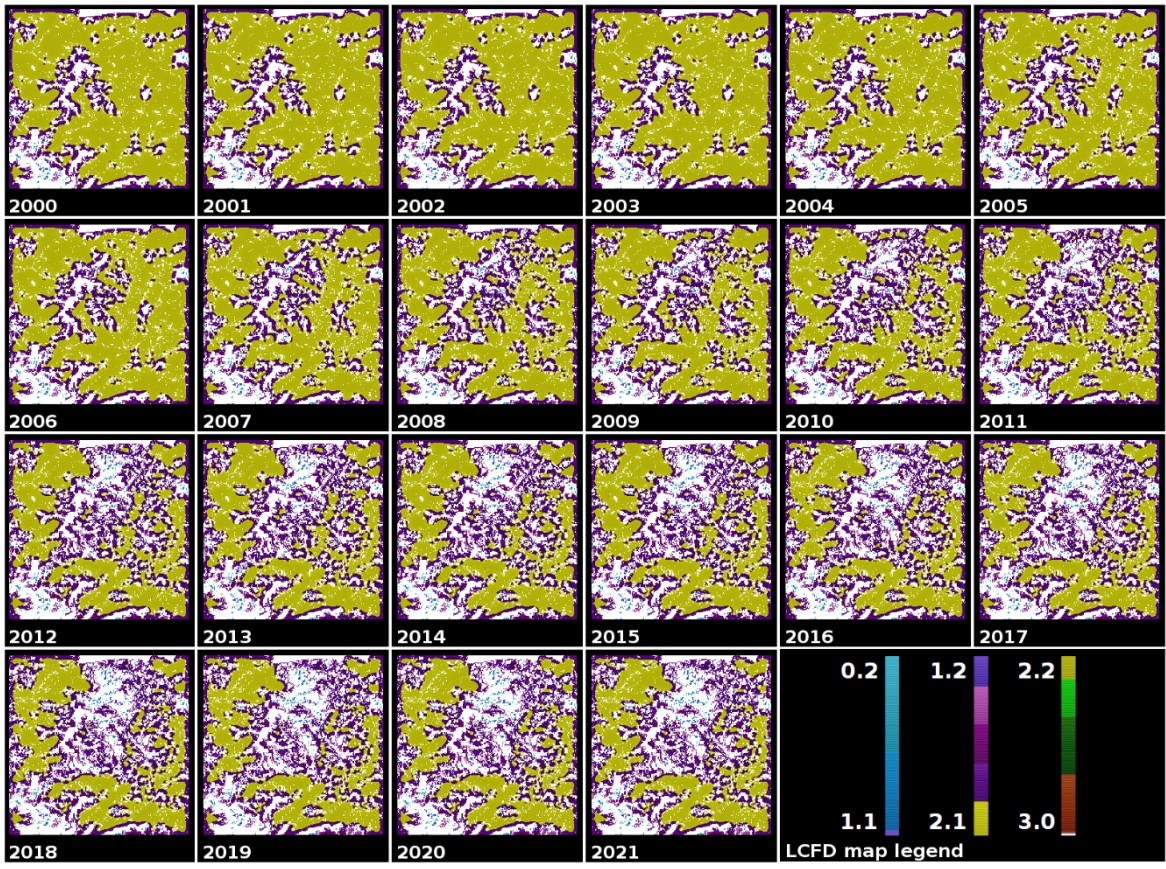

**Figure 10.** The local connectivity maps for the northern group of the Eastern Carpathians from 2000 to 2021. The maps depict maximum connectivity in yellow (*LCFD* = 2), reduced connectivity in purple (*LCFD* between 1 and 2), and very low connectivity in blue (*LCFD* < 1). Areas without forests or deforested zones are shown in white. The maps highlight the reduction in connectivity due to deforestation activities over the specified period.

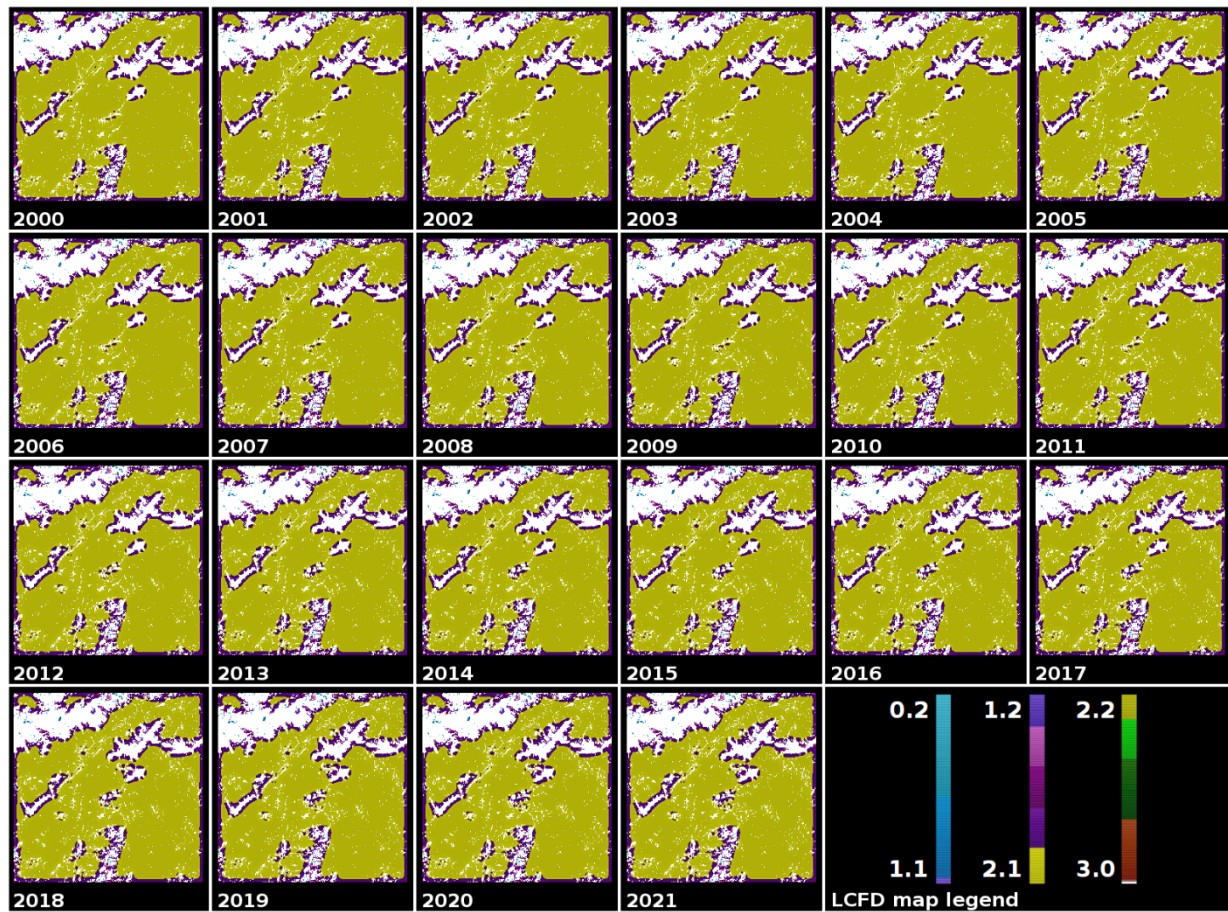

**Figure 11.** The local connectivity maps for the Retezat–Godeanu mountain group from 2000 to 2021. The maps depict maximum connectivity in yellow (*LCFD* = 2), reduced connectivity in purple (*LCFD* between 1 and 2), and very low connectivity in blue (*LCFD* < 1). Areas without forests or deforested zones are shown in white. Compared to Figure 10, the smaller-scale deforestation activities resulted in a slight reduction in compactness and connectivity.

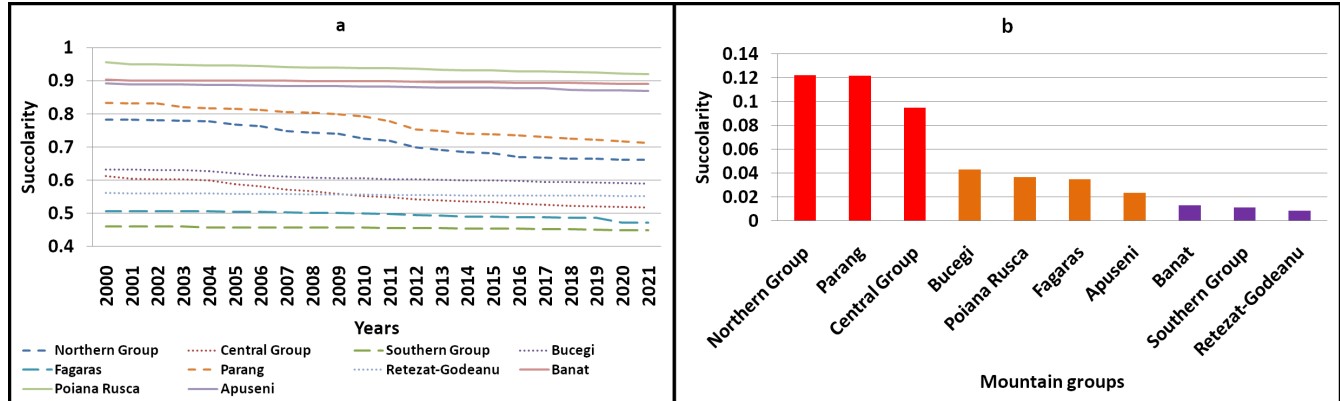

**Figure 12.** Patterns of forest loss effects on tree cover in the 10 mountain groups of the Romanian Carpathians, identified via the analysis of (**a**). mean succolarity and (**b**). differences in succolarity between 2000 and 2021.

Succolarity analyses were conducted for the four directions of tree cover across the 10 mountain groups. The mean succolarity patterns and differences between 2000 and 2021 were revealed (Figures 13a–d and 14a–d). In the *r2l* direction (Figure 13d), the central group has a succolarity value close to 0 because there is a barrier at the right edge of the

image that lacks forests. Figure 14a–d show that the 10 groups are relatively isotropic, with differences in succolarity between 2000 and 2011 for the four directions. Notable variations are only observed in the *r2l* direction, where the central group has a succolarity of 0 (Figure 13d), indicating no connectivity loss due to deforestation, resulting in relatively discontinuous forest patches. Figure 13e,f highlight the spatial isotropy of forests across the 10 mountain groups and their potential and Δ succolarity. The central group is the only group that exhibits succolarity anisotropy, with the *r2l* direction being the differentiator. The analysis of forest area compactness and connectivity dynamics was complemented by the evaluation of fractal indices (*FFI*, *FFDI*, and *LCFD*).

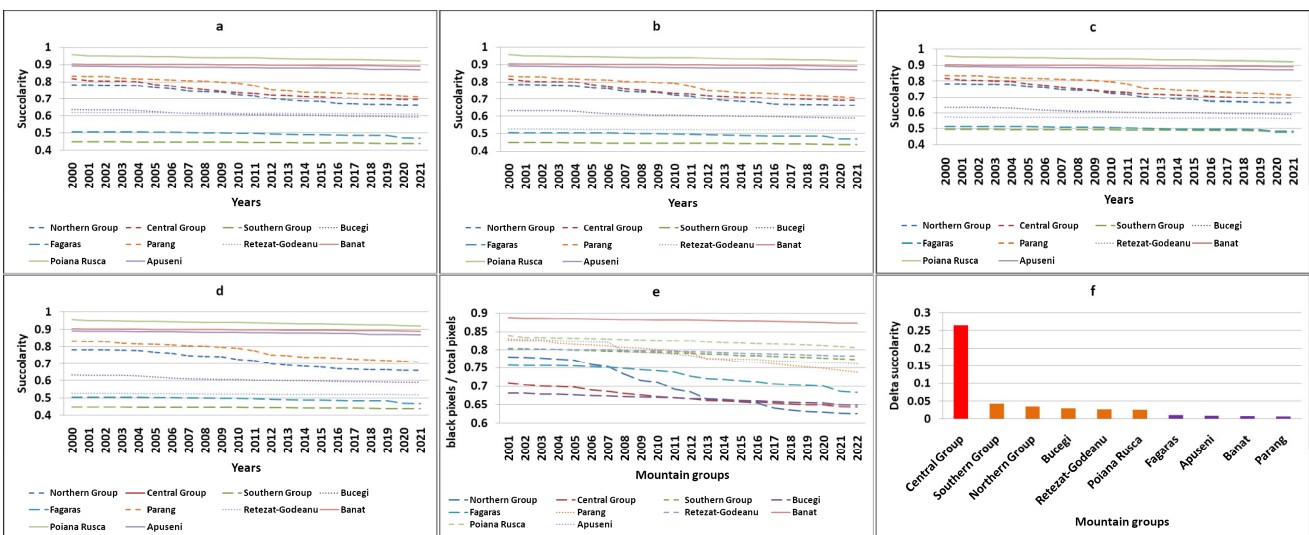

**Figure 13.** Patterns of forest loss effects on tree cover in the 10 mountain groups of the Romanian Carpathians, depending on the direction of analysis, identified by means of an analysis of succolarity: (**a**). top-to-down; (**b**). down-to-top; (**c**). left-to-right; (**d**). right-to-left; (**e**). potential succolarity; and (**f**). Δ succolarity.

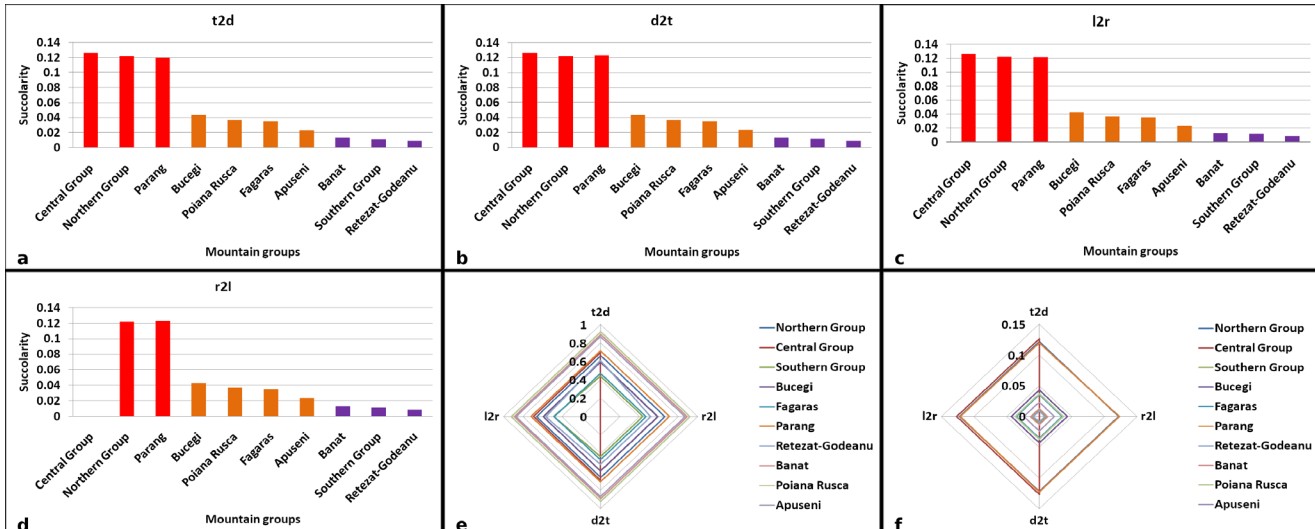

**Figure 14.** Patterns of forest loss effects on tree cover in the 10 mountain groups of the Romanian Carpathians, depending on the direction of analysis, identified by means of an analysis of differences in succolarity between 2000 and 2021: (**a**). top-to-down; (**b**). down-to-top; (**c**). left-to-right; (**d**). right-to-left; (**e**). potential succolarity; and (**f**). Δ succolarity.

Despite increased fragmentation and disorder (*FFI*, *FFDI*) and reduced local connectivity (*LCFD*) in the northern and central groups for the years 2020–2021, these groups still fall

within the category of moderate or average succolarity (Figure 15). The presence of natural or deforestation-induced obstacles limits connectivity among forest patches, leading to fragmentation. Succolarity is an effective measure of the impact of connectivity disruption on fragmentation/compactness, as shown by the varied values between fractal parameters and succolarity. The Banat group is the most compact and connected, with a succolarity around 0.09, which is higher than that of the more fragmented Poiana Rusca group. Despite being less compact, the latter exhibits excellent interconnectivity among forest fragments.

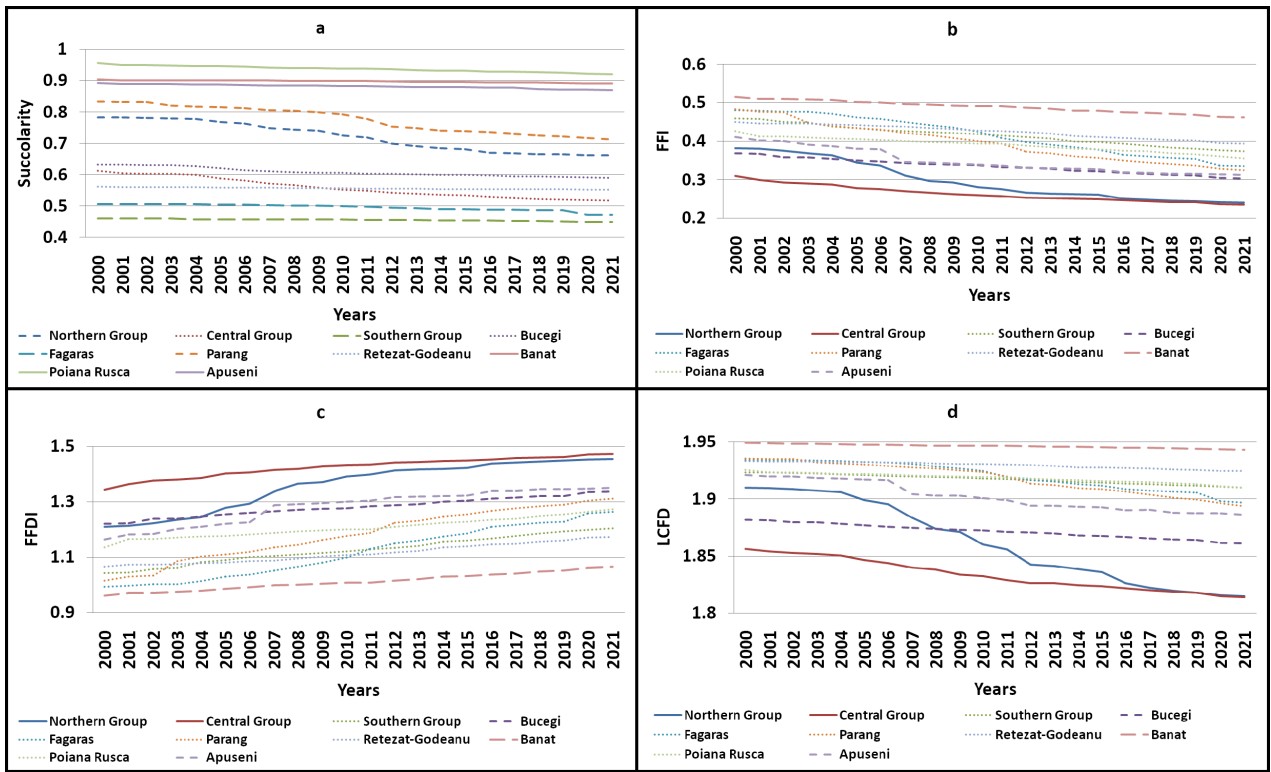

**Figure 15.** Patterns of forest loss effects on tree cover in the 10 mountain groups of the Romanian Carpathians identified via an analysis of the following: (**a**). succolarity; (**b**). *FFI*; (**c**). *FFDI*; and (**d**). *LCFD*.

In conclusion, the metrics of succolarity, connectivity, and fractality reveal the complex dynamics of forested areas in the Romanian Carpathians. This highlights the need for sustainable management practices to preserve the integrity of these invaluable ecosystems.

## 4. Discussion

The analysis of fragmentation and connectivity is crucial for several key reasons, as it provides essential information for biodiversity conservation, sustainable ecosystem management, and understanding the impact of human activities on the environment.

### 4.1. Succolarity and Fractal Indices

This section provides a detailed explanation of the results of our succolarity analysis, highlighting how the size and connectivity of the black patches have a significant impact on the values obtained.

This study uses test images from Figures 1 and 2 to explore the potential of succolarity in quantifying fragmentation/compactness and connectivity. Furthermore, the results are juxtaposed with three fractal indices—*FFI*, *FFDI*, and *LCFD*—for a comprehensive understanding. These results are used as a basis for investigating the impact of deforestation on forest fragment density and connectivity over a 22-year period in ten mountain regions of the Romanian Carpathians.

Succolarity is a versatile tool for percolation analyses, especially in scenarios where the focus is on understanding how substances can traverse diverse pixel configurations. Succolarity, as a measure of percolation, quantifies only the capacity of a virtual liquid to flow through an image (black pixels), circumventing or being blocked by obstacles (white pixels).

The measurement of succolarity provides a concise assessment of permeability and connectivity, capturing variations in the virtual water flow across images and yielding valuable insights into structural properties. This metric is advantageous because it can distinguish between isotropic and anisotropic patterns [82,83] and functions as a robust tool for delineating the effects of wettability on flow capacity [84]. In the complex study of pore systems, succolarity emerges as a key method, traditionally used to characterise structural information [83,85].

The utility of succolarity transcends disciplines, with Ayad and Bakkali's (2019) study demonstrating its effectiveness in identifying disturbances in the geoelectric images of a given area [86]. This correlation between disturbance rates and fractal values is particularly valuable in delineating phosphate deposits with different levels of risk. Similarly, Ref. [87] found a robust correlation between succolarity and permeability in natural gas hydrate samples.

Biological and medical applications of succolarity are exemplified by studies such as [88–90], which provide accurate descriptions of porosity, trabecular structural patterns, and vascular variations. Talu et al., (2020) investigated nanoscale topography using succolarity, highlighting its consistency in indicating high percolation [91].

Mah et al.'s, (2022) investigation of myenteric plexus interstitial cells of Cajal networks highlights the directional and regional variations captured by means of a succolarity analysis across mouse antrum regions [92].

Analyses conducted on test images (Figures 1 and 2) have shown that it is possible to quantify those areas beyond obstacles that can be percolated if the obstacles can be penetrated, by introducing two new indices: potential succolarity (indicating how much the succolarity could increase if virtual percolating water managed to penetrate obstacles) and Δ succolarity (quantifying those pixels beyond obstacles that can still be included in percolation). The results obtained using potential succolarity and Δ succolarity provide a unique perspective on percolation and permeability. In situations where the succolarity is zero, these new concepts are relevant for distinguishing between a completely white image and an image with white margins that prevent virtual water from penetrating in certain directions. Potential succolarity and Δ succolarity emphasize the significance of "inactive" black pixels in the context of a white barrier or ruptured connections. This enhances our understanding of influences on the permeability of the analysed images. The concepts contribute to exploring the relationship between succolarity and permeability, improving comprehension of the results.

However, succolarity does not replace *FFI* or *FFDI* because it quantifies percolation through fragmentation/compaction in accordance with the presence or absence of connectivity, without quantifying the shape of objects (*FFI*) or spatial disorder (*FFDI*). Thus, if it is necessary to determine fragmentation or compaction in accordance with the shape of objects in images, *FFI* remains the fractal solution, and, if spatial disorder is also desired, *FFDI* remains a fractal solution.

The study of connectivity in fields as diverse as medical imaging, ecology, and urban planning has been greatly enhanced by the use of the *LCFD*. This index, derived from fractal analysis, serves as a valuable tool for quantifying the local connectivity of structures or networks, providing insights into the intricate complexities and interconnectivities within the systems being analysed. Originating from the work of Voss and Wyatt (1993) [53], who suggested its efficacy in aligning with human visual perception, *LCFD* has since found applications in diverse domains, providing a nuanced understanding of local connectivity within a broader framework. *LCFD* has been used in various domains. It has been applied to analyse digitised angiograms, revealing distinctions between normal and pathological retinas [79]. Furthermore, it has quantified local complexity at the epithelial–connective

tissue interface, highlighting irregular regions in histological specimens [93]. In medical contexts, *LCFD* has identified circular configurations and increased density in blood vessels associated with female genital schistosomiasis [94]. Its effectiveness has been demonstrated in characterising the complexity of angiogenesis, with higher *LCFD* values corresponding to intricate processes of sprouting and fission in collateral vessels [95]. *LCFD* has explored changes in fractality in mouse embryonic lenses, correlating them with spatial protein expression patterns [96]. In urban planning, *LCFD* has effectively mapped urban connectivity, isolating areas with different connectivity dimensions [97]. The first study in which the local connected fractal dimension was applied in forestry was the analysis of the forests of the Apuseni Mountains [57]. The *LCFD* method has also been used to analyse deforestation processes in the Amazon. In the latter, it revealed differences between isolated patches and more complex connected regions [98].

Thus, the *LCFD* is a powerful analytical tool that crosses disciplinary boundaries. From detecting retinal pathology to dissecting urban connectivity and unravelling the complexity of forest ecosystems, the *LCFD* has proven its worth. Its ability to capture nuanced local interrelationships makes it an invaluable asset in diverse scientific endeavours and promises to continue to contribute to our understanding of complex systems and patterns.

Fractal fragmentation indices serve as valuable tools for assessing the impacts of deforestation at the regional and other spatial scales, providing insights into the complex dynamics of forest fragmentation. The *FFI* has been widely used to analyse the impact of deforestation at various scales. However, the *FFDI* is an improved index that incorporates the Rényi information dimension to provide a more comprehensive assessment of both fragmentation and spatial disorder within images. Despite their utility, these indices, which rely on a box-counting analysis, have a significant limitation—their sensitivity to an image's scale [81]. The sensitivity of the *FFI* and the *FFDI* to image size presents a challenge, making results incomparable when analysing images of different scales.

In summary, succolarity has allowed the identification of directional differentiation and the role of barriers in the connectivity of natural or deforestation-induced forest fragments. Unlike succolarity, the *FFI*, the *FFDI*, and the *LCFD* are non-directional analyses and give identical results regardless of an image's rotation (see Figure 7d–f). The *LCFD* has been shown to be insensitive to the presence of obstructions that disrupt connectivity.

The *FFI* only identifies the degree of fragmentation and compaction, whereas forest patches may or may not be connected, allowing or hindering connectivity. Similarly, the *FFDI* has the same limitation as the *FFI* in that it only indicates the general spatial arrangement of forest patches without taking into account barriers or connectivity.

The use of succolarity in this context overcomes the shortcomings or limitations of the three indicators analysed by providing information on fragmentation/compaction and connectivity, taking into account both direction and barriers. This approach provides a more comprehensive understanding of the complex dynamics of forest structures and their connectivity, taking into account directional variations and barriers in a way that the other indices do not.

### 4.2. The Limits of Succolarity

However, succolarity has certain limitations. The first limitation lies in the fact that its analysis is restricted to binary raster forest images, making it impossible to analyse vectorial, RGB, or grey-scale images. As a result, satellite image pre-processing is required, which includes forest extraction and its subsequent binarization. A second limitation arises from the binarization process, which can introduce artefacts. Ensuring an accurate binarization is crucial for reliable results. A third limitation relates to the way in which succolarity quantifies the degree of fragmentation and connectivity. The analysed image should not have forest-free edges (a white border), as shown in Figure 3f,l, because succolarity will automatically register a value of 0, even if there are fragmentation and connectivity beyond this border which could be quantified. To overcome this limitation, it is necessary to remove the white border using the crop function. As shown in Figure 6a, succolarity is not sensitive

to image size, so the loss of an image's edges will not affect the final analysis results, unlike other fractal analyses.

### 4.3. Analysis of Succolarity and Fractal Index in a Forest Analysis

Until now, the *FFI* was used to examine the fractal fragmentation of forests in various contexts. The results of this index showed a consistent trend of increased fractal fragmentation following deforestation [54,57–62,99–101].

This study presents, for the first time, the use of succolarity analysis to investigate and measure the impact of forest loss on the fragmentation, compaction, and connectivity of forested areas. Succolarity is a well-known method for analysing pore systems, characterizing permeability, and quantifying infiltration rates. However, it has the potential to provide valuable insights into the complex relationship between forest structure and connectivity. Furthermore, this study illustrates the fact that succolarity can be utilised beyond traditional boundaries, broadening the scope of its usefulness and providing a detailed comprehension of the diverse impacts of deforestation on forested areas.

More than that, this study provides compelling evidence for the viability of succolarity analysis and offers improved solutions for quantifying forest fragmentation, compaction, and connectivity in different directions. In this innovative approach, barriers are defined as either forested or deforested areas, while areas which allow infiltration are treated as being forested. The nuanced patterns revealed following deforestation highlight the importance of succolarity in capturing complex variations in forest structure.

This study demonstrates that fractal dimensions such as the *FFI*, the *FFDI*, and the *LCDF* are sensitive to image size, which can introduce biases in compactness analyses. To ensure comparability, any image analysis conducted on different images must be performed at the same scale.

In contrast, succolarity provides consistent values across different image scales. Succolarity maintains identical values from $64 \times 64$ to $1024 \times 1024$ pixels, indicating its invariance and robustness as a metric. This study offers insights into the reliability of succolarity as a metric for assessing permeability and connectivity in images of varying scales. This metric is advantageous because it can distinguish between isotropic and anisotropic patterns [82] and assess the potential for percolation. It captures variations in the virtual water flow across images, providing valuable insight into structural properties. In the case of the analysis of forest images, it provides information about the spatial effects generated by deforestation and the latter's impact on the continuity of habitats.

Three distinct patterns were revealed in the decrease in succolarity following deforestation and natural forest loss, as shown in Figures 8b and 13a–d. However, these patterns were not uniform across all areas. This study showed that extensive clear-cutting of forests in the northern group [61,62], in the central group [59], and even in the Parang group [66] mainly resulted in the fragmentation of compact forest areas and the disruption of connectivity. This led to a decrease of approximately 0.12 in succolarity over a 22-year study period. Deforestation has significantly reduced and fragmented the forests in the Carpathian Mountains [39]. As a result, many regions have become too small to support viable populations of species such as bears [102–104], wolves [105], or capercaillie [106] due to the increasing isolation of larger forested areas with suitable habitats. Therefore, the measurement of succolarity, as well as potential succolarity and Δ succolarity, can be a valuable tool in identifying areas where connectivity can be restored. This, in turn, allows for habitat recreation through reforestation.

Succolarity is a metric that measures the loss of connectivity between forest patches due to fragmentation. It improves upon existing indices by using metrics that account for directionality when quantifying fragmentation and connectivity. Analysing both fragmentation and connectivity is crucial for assessing the evolution of forest ecosystems. The obtained results are valuable for biodiversity conservation and ecological restoration.

A fragmentation and connectivity index, such as succolarity, can make a significant contribution to biodiversity conservation and ecological restoration by providing essential

information and guidance for the sustainable management of forest ecosystems. Succolarity identifies critical areas where fragmentation is high and connectivity is low, providing guidance for focusing conservation efforts on those areas with the greatest impact. These data can also be used to prioritize conservation actions. In addition, the planning of ecological corridors to restore connectivity between fragmented forest patches can be guided by potential and delta succolarity measures.

Succolarity can also be used to assess changes in fragmentation and connectivity over different time periods, facilitating species movement and maintaining healthy genetic flow to strengthen biodiversity. This text provides a basis for monitoring the success of conservation and ecological restoration efforts and for adjusting strategies in real time. Succolarity can offer a comprehensive view of the condition of forest ecosystems, aiding in sustainable management, conservation of biodiversity, and restoration of ecology efforts.

### *4.4. Future Work*

Future work should primarily focus on extending anisotropy analyses based on succolarity [82]. This includes identifying areas where the connectivity and consolidation of forest areas can be increased through networking, particularly afforestation efforts. This approach can facilitate both biodiversity conservation and ecological restoration. In addition, there is a need for studies that apply the current methodology to other forest areas affected by deforestation in different climatic zones (tropical forests, taiga) to ensure the adaptability and effectiveness of this approach in different ecosystems.

### 5. Conclusions

Succolarity is a versatile tool for percolation analyses, especially in scenarios where the focus is on understanding how substances can traverse diverse pixel configurations.

Succolarity emerges as an outstanding analytical tool in the study of Carpathian forests, offering advantages in the quantification of fragmentation, connectivity, and permeability. Its ability to discriminate between isotropic and anisotropic patterns, coupled with directional precision and invariance to image scale, makes it effective. This study highlights the robustness of succolarity in contrast to the sensitivity of the *FFI* and the *FFDI* to image scale, reinforcing succolarity as a fundamental measure for forest analysis. The integration of succolarity with traditional indices improves the overall understanding of complex forest dynamics. The application of succolarity in the Carpathians not only reveals new insights into structural evolution but also highlights its adaptability to different changes. This study demonstrates the reliability of succolarity in capturing intricate details of deforestation-induced connectivity disruption, providing valuable guidance for sustainable forest management.

In conclusion, our extensive research on succolarity, the local connected fractal dimension (*LCFD*), the fractal fragmentation index (*FFI*), and the fractal fragmentation and disorder index (*FFDI*) has not only deepened our understanding of structural complexity, permeability analysis, and forest fragmentation but has also introduced novel indices (potential and $\Delta$ succolarity) that promise increased precision in assessing environmental phenomena. As a method with proven efficacy in geological, biological, and medical fields, succolarity is a versatile analytical tool with the potential to identify intricate relationships within diverse systems.

**Funding:** This research received no external funding.

**Data Availability Statement:** All the data generated or analysed during this study are included in this published article in the form of figures. Any additional information about the dataset or the dataset in a different format than what is presented in this article can be obtained from the corresponding author upon request.

**Acknowledgments:** I.A. would like to thank Distinguished Ioannis Liritzis for his valuable suggestions and comments, which helped the author to improve the article. I.A. is thankful for the support of the PNCDI III, the grant of the Ministry of Research, Innovation and Digitization, CNCS/CCCDI-UEFISCDI, with project number PN-III-P2-2.1-SOL-2021-0084.

**Conflicts of Interest:** The author declares that they have no competing financial interests or personal relationships that could have influenced the work reported in this paper.

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
