# Peer review of "Analysis of Forest Fragmentation and Connectivity Using Fractal Dimension and Succolarity"

_land, doi:10.3390/land13020138_

Round 1

Reviewer 1 Report

Comments and Suggestions for Authors

General comments

The research presents an in-depth investigation of forest fragmentation and connectivity in the Romanian Carpathians, utilizing the notion of succolarity. The introduction of novel indices, such as potential succolarity and delta succolarity, allows for the evaluation of percolation and landscape dynamics. The study used fractal analytic techniques, such as succolarity, Fractal Fragmentation Index (FFI), Fractal Fragmentation and Disorder Index (FFDI), and Local Connected Fractal Dimension (LCFD), to investigate the effects of deforestation over a span of 22 years. The research highlights the efficacy of succolarity in capturing complex patterns of forest structure and connectivity, providing valuable insights for the promotion of sustainable forest management and conservation endeavors.

Introduction

Lines 37 - 49. These two paragraphs could be merged into one paragraph since they are essentially focusing on the impacts of deforestation and habitat fragmentation. You could connect the two paragraphs using ‘For instance’ and then delve into the impacts you have discussed.

Line 50 - 57: These two paragraphs are too short and related and therefore can be merged into one paragraph once more.

Materials and methods

Line 159: written ‘the can’ instead of ‘there can’

General review on methods

While the study introduces an innovative methodology, nevertheless, it is important to acknowledge its potential constraints. Notably, the computational intricacy and data prerequisites for conducting such meticulous studies may not be practical in every study setting. Further investigation could be conducted to assess the efficacy of the study in practical scenarios and its versatility in various forest environments.

Comments on the Quality of English Language

The English language quality is fair. A few grammatical errors have been noticed but overall acceptable.

Author Response

Thank you for reviewing the manuscript. Your suggestions were very valuable and helpful.

Reviewer 2 Report

Comments and Suggestions for Authors

General Comments:
The manuscript presents a comprehensive analysis of forest fragmentation and connectivity in the Romanian Carpathians using fractal dimension and succolarity. The study proposes two novel indices, potential succolarity and delta succolarity, to provide insights into environmental changes and human interventions in forests. The manuscript provides detailed explanations of the methodology and results. Overall, the study is valuable and contributes to the understanding of forest fragmentation and connectivity. However, there are some suggestions for improvement and clarification.

Specific Comments:

1.The introduction provides a clarification for the importance of forest fragmentation and connectivity. However, there is a need to further summarize the existing problems in the current quantification methods of forest fragmentation and discuss the advantages and disadvantages of various methods. Why is the Succolarity index necessary for assessing forest fragmentation, and how can the Succolarity index along with the two novel indices proposed by the author solve specific issues in forest fragmentation monitoring? Additionally, it is suggested to provide necessary elaboration on the typicity and representativeness of choosing the Romanian Carpathians as the research subject.

2.The methodology section is well-documented and provides sufficient information to understand the analysis process. However, I suggest that the Succolarity method could be described more concisely by incorporating additional illustrations, as the quantification of Succolarity has already been adopted in previous studies.

3.The results section presents the findings of the study effectively and provides detailed insights into forest fragmentation and connectivity in the Romanian Carpathians. However, the figures in this section should be aesthetically pleasing and provide more comprehensive information to meet the publication standards of academic journals. Additionally, it would be helpful to include more visual representations, such as maps showing fragmentation in your study site, to illustrate the patterns and trends observed during 2001-2021.

4.I believe that the discussion section should further elaborate on the implications of the results and their significance in the field of forest fragmentation, such as biodiversity conservation and landscape management, among others. In the current version of the manuscript, the substantive application of Succolarity in relation to forest fragmentation is clearly insufficient. It is suggested to provide more explanations regarding the significance of these results in the context of the Romanian Carpathians. Furthermore, it would be valuable to include recommendations or future research directions based on the results. This could help readers understand the practical applications of the study and identify potential areas for further investigation.

5.In line 297, is there a missing period after "by Plotnick"?

6.Remove the hyphen from "pix-els" in row 396.

7.Figure 3 should be produced with higher quality to ensure sufficient clarity.

Author Response

(The authors gave the same response as above.)

Reviewer 3 Report

Comments and Suggestions for Authors

The manuscript devoted to analysis of forest structure in the Romanian Carpathians using traditional and new developed indices. The author applies the succolarity index for study of the satellite images for the first time. The main idea of the paper is to estimate the forest structure in the terms of the total “connectivity”. The suggested approach is original and will be interesting for readers.

Author develop new indexes (succolarity; potential succolarity; delta succolarity) for describing forest structure and their changes. These indexes were compared to other indexes used for forest fragmentation analysis. The advantages of new indexes in comparison to old ones is in quantification of the fragmentation and connectivity of forest patches.

According to author, tree cover is defined as areas where virtual water can percolate (line 79), and areas without forests are considered obstacles that force the virtual water to bypass or be blocked. Is seems strange from physical point of view. Wood is denser than air. Likely trees should be considered as obstacles. This notation does not change the mathematical meaning of the used approach, but can be mentioned in the introduction.

The methods section is well written and describes used approaches and parameters. Methods were tested on artificially generated images to access the sensitivity of new indexes. After validation, new indexes were applied for investigation of forest cover dynamics over ten mountain groups of the Romanian Carpathians.

Results and Discussion are clear and supported by Conclusions. Conclusions contain a summary of advantages of new indexes in comparison with old ones.

References are appropriate and contain a large list of relevant papers.

Figures are good and described within the text. The suggested method is quite new and will be a good tool for ecosystem studies.

Specific comments.

Lines 139, 145, 169, 175, 194, 227… Equations should be black.

Lines 630-638 – looks like a part from introduction. Remove of rephrase.

Author Response

(The authors gave the same response as above.)
